# Multifaceted conserved functions of Notch during post-embryonic neurogenesis in the annelid *Platynereis*

Loïc Bideau [iD][1], Loeiza Baduel [iD][1], Gabriel Krasovec [iD][1], Caroline Dalle[1], Ombeline Lamer [iD][1], Mélusine Nicolas[1], Alexandre Couëtoux[1], Corinne Blugeon[2], Louis Paré [iD][1], Michel Vervoort[1,3], Pierre Kerner [iD][1] & Eve Gazave [iD][1✉]

## Abstract

Notch signaling is an evolutionarily conserved pathway known to orchestrate neurogenesis by regulating the transition from progenitors to neurons and glia, and by directing neurite outgrowth and axon guidance in many species. Although extensively studied in vertebrates and ecdysozoans, the role of Notch in spiralians remains unexplored, limiting our understanding of its conserved functions across bilaterians. Here we focus on the segmented annelid *Platynereis dumerilii*, a model organism in neurobiology and regeneration, to investigate Notch signaling functions during post-embryonic developmental processes. We show that Notch pathway components are expressed in neurogenic territories during both posterior elongation and regeneration, two processes requiring sustained neurogenesis. Through chemical inhibitions of the pathway and RNA-seq profiling, we find that Notch signaling regulates neural progenitor specification, differentiation, and overall neurogenic balance in the regenerating and elongating posterior part. Disruption of Notch signaling activity leads to severe defects in pygidial and central nervous system organization. Altogether, our results support the hypothesis that Notch has multifaceted conserved functions in neurogenesis across bilaterians, shedding light on the ancestral functions of this critical pathway.

**Keywords** Annelid; Neurogenesis; Notch; Posterior Growth; Regeneration
**Subject Categories** Development; Evolution & Ecology; Neuroscience

## Introduction

Notch is an ancient signaling pathway, probably already functional in the last common ancestor of Metazoa (Gazave et al, 2009; Lv et al, 2024), which modulates a large array of cell fate decisions in a variety of developmental processes (Bray, 2006; Henrique and Schweisguth, 2019). Notch is a membrane receptor that is cleaved upon binding the ligand Delta/Jagged, located on a neighboring cell. Thus, Notch is a contact-dependent (or juxtacrine) signaling pathway (Andersson et al, 2011; Gazave et al, 2009; Gazave and Renard, 2010; Hori et al, 2013). After cleavage, its intracellular domain is translocated into the nucleus of the receiving cell where it interacts with a transcriptional complex that regulates a large number of genes, including members of the Hairy Enhancer of Split (or Hes) superfamily (Gazave et al, 2014; Kageyama et al, 2007) (Appendix Fig. S1A).

Notch is renowned for its fine-tuned regulation of cellular transitions from a progenitor to a differentiated state through a process known as lateral inhibition, whereby a cell inhibits its neighboring cells from adopting the same fate (Cau and Blader, 2009; Henrique and Schweisguth, 2019; Sjoqvist and Andersson, 2019). This is especially crucial during neurogenesis, where lateral inhibition, through Notch signaling, is key for regulating the transition from neural progenitor cells to mature neural cells. In short, Notch signaling is activated among a cluster of proneural cells where it specifies a neural precursor cell to become either a glial or a neuronal cell. This process maintains an essential balance for proper nervous system development in different animals such as *Drosophila* (Bahrampour and Thor, 2020; Henrique and Schweisguth, 2019; Sood et al, 2022) or vertebrates (Chouly and Bally-Cuif, 2024; Pierfelice et al, 2011). In both lineages, the inhibition of Notch signaling triggers similar neurogenic phenotypes consisting in an excess of neural cells (Pierfelice et al, 2011). In addition, Notch regulates axon guidance and neurite outgrowth in both *Drosophila* (Kannan et al, 2018; Kuzina et al, 2011; Zhang et al, 2023) and vertebrates (Aujla et al, 2011; Shi et al, 2011) during embryonic neuronal maturation. As neurogenesis is a lifelong

[1]Université Paris Cité, CNRS, Institut Jacques Monod, F-75013 Paris, France. [2]GenomiqueENS, Institut de Biologie de l'ENS (IBENS), Département de biologie, École normale supérieure, CNRS, INSERM, Université PSL, Paris, France. [3]Deceased: Michel Vervoort. ✉E-mail: eve.gazave@ijm.fr

process, Notch signaling is also involved during homeostatic adult neurogenesis (Chouly and Bally-Cuif, 2024; Lampada and Taylor, 2023). While data from vertebrates and *Drosophila* may suggest conserved roles of Notch in neurogenesis, the scarcity of functional analyses in the third large bilaterian lineage, the spiralians, prevents from drawing definitive conclusions regarding the evolution of its functions at the bilaterian scale (Lv et al, 2024; Morizet et al, 2024).

Among spiralians, the segmented annelid *Platynereis dumerilii* has emerged as a powerful model for diverse research fields such as development, evolution, regeneration, and neurobiology (Ozpolat et al, 2021; Schenkelaars and Gazave, 2021). Its complex larval nervous system encompasses both a central nervous system (CNS) and a peripheral nervous system (PNS). CNS consists of an anterior brain, a ventral midline separating a stratified neuroectoderm and a bona fide ventral nerve cord (VNC). The peripheral nervous system is connected to the CNS and is mainly composed of ganglions innervating the appendages (parapodia) and their sensory structures. *Platynereis'* larval trunk neuroectoderm is organized through proliferative/differentiating apico-basal layers and medio-lateral domains, prefiguring longitudinal tracks of different neuron types. This organization highlights key evolutionary conserved processes, such as medio-lateral patterning of the neuroectoderm (Denes et al, 2007) or the conservation of proneural bHLH gene functions among bilaterians (Simionato et al, 2008). Previous studies identified the core members of the Notch pathway (*Notch, Delta, Jagged, DSL1-3, Nrarp, Presenilin, Su(H), Numb, Fringe*) as well as its putative target *Hes* genes, which are members of the bHLH family, constituting a complete Notch repertoire in *Platynereis* (Gazave et al, 2014; Gazave et al, 2017). These genes were shown to be involved during embryogenesis in the correct patterning of the structure producing extracellular bristles or chaetae, the chaetal sac, likely through a lateral inhibition mechanism. Yet, no major role during embryonic or early larval neurogenesis was observed (Gazave et al, 2017). This prompted us to investigate the function(s) of Notch pathway during two key post-embryonic developmental processes in *Platynereis* - posterior elongation and posterior regeneration - which both require sustained neurogenesis to innervate the newly formed tissues (Gazave et al, 2013; Planques et al, 2019). Juveniles elongate by the addition of newly formed segments at their posterior end, thanks to a growth zone (GZ, also called segment addition zone) of active progenitors or stem cells, located right above the terminal part of the worm bearing the anus, the pygidium (Gazave et al, 2013). Juveniles have also the ability to replace a lost or injured posterior part, including the GZ and the pygidium (Poss, 2010), through restorative regeneration (Bely and Nyberg, 2010; Bideau et al, 2021) (Appendix Fig. S1B). Similar to most of regeneration processes, *Platynereis'* posterior regeneration can be divided into three common and sequential steps (Tiozzo and Copley, 2015). First, a wound healing closes the wound and produces a wound epithelium. The second step usually relies on the formation of a regeneration-specific structure called a blastema, following the mobilization of precursor cells. Third, late morphogenetic processes involving patterning, differentiation and growth of the reformed structure constitute the final step (Bideau et al, 2021; Galliot and Ghila, 2010; Tiozzo and Copley, 2015).

Here, we showed that the core members of the Notch pathway are dynamically expressed in all neurogenic territories throughout all steps of regeneration as well as during posterior elongation. Thanks to chemical inhibition and differential RNA-seq analyses,

we found that Notch pathway, potentially through the action of the ligand *Delta* and several *Hes* genes regulates both the determination of neural progenitors and the balance of differentiated neurons in the regenerated pygidium. During posterior elongation, Notch inhibition disturbs the neurogenic cascade dynamics inducing a thickened CNS neuroectoderm and an abnormally-shaped VNC with neurite growth defects. Altogether, our study supports the idea that Notch plays multiple and conserved roles during adult neurogenesis in bilaterians.

# Results

## Dynamic expression of Notch pathway components and target genes in neurogenic structures during *Platynereis'* posterior regeneration

We have already determined that the genome of *Platynereis* contains the core components of the Notch pathway, including receptor *Notch*, two canonical ligands (*Delta* and *Jagged*), a series of putative alternative ligands (*Delta/Serrate-like* or *DSL* genes), as well as the gamma-secretase machinery and the negative regulator *Nrarp* (Gazave et al, 2017). Additionally, we previously characterized the *Hes* gene superfamily, identifying 15 members as putative Notch targets (Gazave et al, 2014; Gazave et al, 2017). In order to evaluate the role(s) of the Notch pathway in the context of posterior regeneration in *Platynereis*, we first analyzed the expression of its ligands, receptor, regulator, member of the gamma-secretase machinery and downstream targets. To this end, we used our previously generated RNA-seq dataset which contains five stages of posterior regeneration as well as the non-amputated structure. Together, these stages encompass the entire regeneration process (Figs. 1A and EV1A; Dataset EV1) (Paré et al, 2023). Most Notch-related genes displayed a dynamic expression, with upregulation peaking at later regenerative stages (from 3 dpa—days post-amputation—onward), suggesting a role in blastema growth and differentiation (Figs. 1A and EV1A; Dataset EV1).

We then assessed their expression patterns during posterior regeneration using whole mount in situ hybridizations (WMISH) thanks to an established staging system covering all main regeneration steps: wound healing, blastema formation and morphogenesis (Planques et al, 2019) (Figs. 1B and EV1B). We observed that among the seven core members of the Notch pathway (*Notch, Delta, Nrarp, DSL1* to *3, Presenilin*), five of them (*Notch, Delta, DSL1, DSL3* and *Presenilin*) were expressed in one or more neurogenic territories i.e. neuroectoderm, ventral nerve cord (VNC), peripheral nervous system (PNS) and sensory pygidial cirri. Similarly, seven out of the twelve *Hes*-related genes were expressed in a nervous system-related structures (Fig. 1B,C).

### Wound healing
Immediately after amputation, most of the genes were not detected in the tissues proximate to the wound with the noticeable exceptions of *DSL1 & 3*, expressed in the VNC of the non-amputated (NA) tissues (Fig. 1B, stage 0). At 1 dpa, the receptor *Notch* was widely expressed in the wound epithelium, while the ligands *Delta, DSL1* and *DLS3* as well as *Presenilin* and some *Hes* genes (*Hes4, 6, 13*) had a more discrete expression in few epithelial cells (Fig. 1B, stage 1).

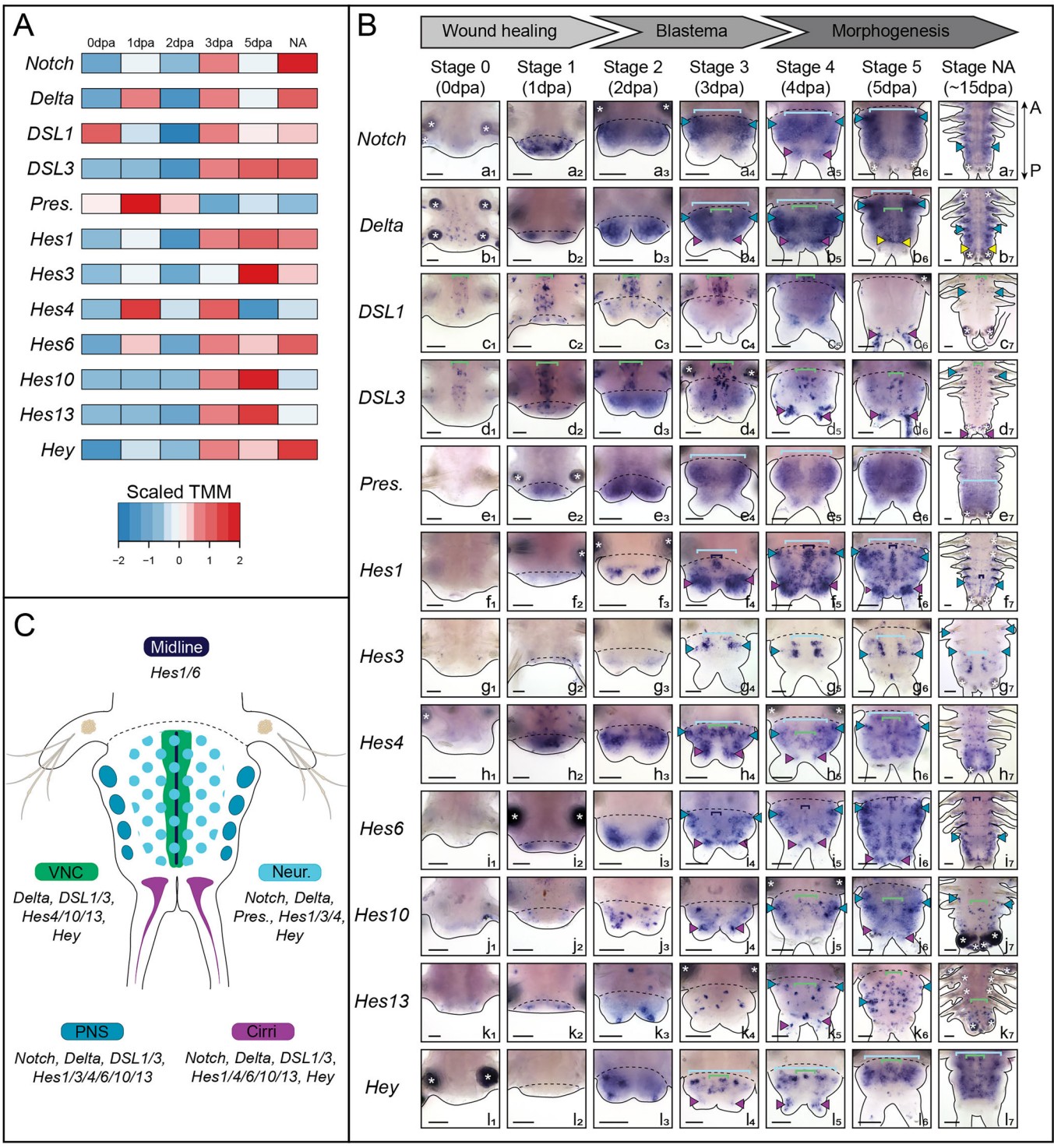

### Blastema formation

At 2 and 3 dpa, *Notch, Delta* and *Presenilin* were broadly expressed in both the mesoderm and the ectoderm, potentially in the neuroectoderm and, for *Notch* and *Delta* in the PNS of the small blastema (Fig. 1B, stages 2 and 3). *DSL1* and *3* were mainly restricted to the VNC of the NA tissues at 2 dpa (Fig. 1B, stage 2). Conversely, *DSL3* expression extended to nerve cells within the

newly regenerated VNC at 3 dpa (Fig. 1B, stage 3). At 2 dpa, some *Hes* genes were expressed in modest (*Hes1, 6*) to large areas (*Hes4*) in the ectoderm of the blastema (Fig. 1B, stage 2). At 3 dpa, these genes exhibited a strong expression in the midline (*Hes1, 6*), pygidial cirri (*Hes1, 4, 6, 10, Hey*), neuroectoderm (*Hes1, 3, 4, Hey*) and the PNS (*Hes3, 4, 6*) (Fig. 1B, stage 3). A number of other genes were expressed in a restricted manner in few scattered ectodermal

**Figure 1. Several core members of the Notch pathway and its putative target genes are dynamically expressed in neurogenic structures during posterior regeneration.**

(A) Heatmap representation of expression levels of several Notch components and *Hes* genes during posterior regeneration (Paré et al, 2023). (B, a to I) Whole-mount in situ hybridizations (ventral views) of Notch components and *Hes* genes expressed in a neurogenic structure during the three main steps of regeneration (wound healing, blastema formation and morphogenesis). (C) Schematic drawing of a regenerated part at stage 5 depicting the expression patterns of the genes found expressed in neurogenic structures during regeneration. Data information: Pres. = Presenilin, VNC = ventral nerve cord (green brackets), PNS = peripheral nervous system (blue arrowheads), Neur. = neuroectoderm (light blue brackets); Cirri (purple arrowheads); Midline (dark blue brackets); yellow arrowheads = growth zone involved in posterior elongation of the animals (Gazave et al, 2013); white asterisks = non-specific staining from parapodial glands. dpa = day(s) post-amputation, NA = non-amputated. Solid black lines delineate the outlines of the samples, black dashed lines correspond to the amputation planes. Scale bars = 50 μm. Anteroposterior (A/P) axis is represented. All images come from representative samples of at least two biological replicates. Source data are available online for this figure.

cells (*Hes10, 13, Hey*), which may be neural cells of the VNC, PNS and/or pygidial cirri at stages 2 and 3.

*Morphogenesis*

At 4, 5 and 15 dpa (a proxy of NA posterior part (Gazave et al, 2013)), all the genes exhibited broadly similar yet larger expression patterns than at stage 3. *Notch*, *Delta* and *Presenilin* were largely expressed in mesodermal and ectodermal tissues including the neuroectoderm, the VNC, the PNS (except *Presenilin*) and the pygidial cirri (Fig. 1B, stages 4, 5, NA). *DSL1* and *3* were expressed in neurons of the VNC and pygidial cirri. *Hes* genes were found in all neural structures: the midline (*Hes1, 6*), the VNC (*Hes4, 10, 13, Hey*), the neuroectoderm (*Hes1, 3, 4, Hey*), the PNS (*Hes1, 4, 6, 10, 13*) and the pygidial cirri (*Hes1, 4, 6, 10, 13, Hey*).

Hence, a combination of at least the *Notch* receptor, one ligand and several *Hes*-related genes are found co-expressed in all neurogenic structures during regeneration (Fig. 1C).

However, the expression of Notch pathway members and *Hes* putative target genes expression is not restricted to neural territories. Instead, expression can be observed in segmental stripes (*DSL2* and *Hes5*, Fig. EV1Ba,d), in the ventral vessel (*DSL2*, Fig. EV1Ba), or in a broad mesodermal area (*Hes8* and *11*, Fig. EV1Be,f). In addition, three genes (*Nrarp, Hes2* and *Hes12*), were expressed in chaetal sacs (Fig. EV1Bb,c,g), i.e. the structures responsible for the production of chaetae composed of follicle cells surrounding a central chaetoblast (as previously described during larval development (Gazave et al, 2014; Gazave et al, 2017)). Finally, *Delta, Nrarp, Hes2* and *Hes8* were expressed in the regenerated growth zone or GZ (Figs. 1B,b and EV1Bb,c,e) (Gazave et al, 2013).

By combining RNA-seq data with the thorough analysis of complex and dynamic expression patterns of the Notch pathway components and putative *Hes* target genes, we hypothesize that Notch orchestrates—albeit not exclusively—key neurogenic functions during posterior regeneration and elongation in *Platynereis*.

## Early Notch pathway inhibition alters neurogenesis and induces pygidial hypertrophy during *Platynereis'* posterior regeneration

We then performed a series of chemical inhibition experiments to determine the role(s) of Notch signaling during the 5-day-long process of posterior regeneration in *Platynereis*. We observed that as early as 2 dpa, the treated worms (exposed to all the tested gamma-secretase inhibitors—LY-411575, RO-4929097 and DAPT—immediately after amputation) displayed a statistically significant delay in regeneration compared to DMSO controls, failing to progress beyond stage 2

(characterized by a small bilobed blastema) (Fig. 2A; Appendix Fig. S2A–C). To determine whether this delay resulted from impaired cell proliferation, which is mandatory for regeneration to proceed (Planques et al, 2019), we performed a 1-h EdU pulse experiment in both DMSO controls and LY-411575-treated worms to label cells in S-phase. At 2 and 5 dpa, the proportion of EdU+ cells were similar between conditions (Fig. 2B, C). However, their distribution differed at 5 dpa. We quantified significantly less EdU+ cells in the endo-mesoderm of the LY-411575-treated worms in comparison to the controls indicating that while proliferation level remained globally alike, some cell types are proliferating differentially (Appendix Fig. S2D). Additionally, a 3-day long EdU chase following a 1-h pulse at 2 dpa, to ensure that cells in S-phase at that stage are indeed dividing, revealed an extensive dilution of the EdU, as well as a characteristic punctuated EdU signal in both control and treated worms, confirming ongoing cell divisions (Appendix Fig. S2E). Thus, the regenerative arrest observed upon Notch pathway inhibition is not due to reduced cell proliferation. Therefore, given the limited growth of the regenerated structure, we determined cell death profile with TUNEL assay (Fig. 2D,E). In DMSO controls, apoptotic cells were detected in internal tissues at 2 dpa, consistent with the expected response to amputation (Vullien et al, 2025), but were nearly absent at 5 dpa. In contrast, LY-411575-treated worms exhibited a significantly higher proportion of TUNEL+ cells at 5 dpa, affecting both internal and superficial tissues (Fig. 2D,E). Thus, Notch pathway inhibition increases cell death during posterior regeneration.

Detailed examination of the morphology and cell nuclei arrangements revealed that regenerated parts from LY-411575-treated worms are not merely arrested at approximately stage 2, but rather exhibit morphological defects. At 5 dpa, the regenerative structure appears hypertrophied, with enlarged and roundish blastemal lobes (Fig. 2F), even though their cellular densities remain unchanged (Appendix Fig. S2F). To further characterize this phenotype, we conducted an extensive molecular analysis using a set of gene markers and labelling experiments to mark different structures and tissues involved in *Platynereis'* posterior regeneration (Figs. 2G,H and EV2) (Gazave et al, 2013; Kostyuchenko et al, 2019; Planques et al, 2019). Ring-like expression of *Hox3* revealed that the ectodermal GZ is maintained upon Notch pathway inhibition (Fig. EV2A) but is positioned more anteriorly in the 5 dpa LY-411575-treated regenerated parts (Fig. EV2A'2) than in the control (Fig. EV2A'1). Similarly, the expression patterns of *Evx, PiwiB, Myc* and *Nanos*, confirmed the presence of a mesodermal GZ in treated worms (Fig. EV2B–E). However, these markers also indicated a massive reduction in the territories occupied by mesodermal progenitors within the blastema (Fig. EV2B–E). Moreover, the typical striped expressions of the segmentation

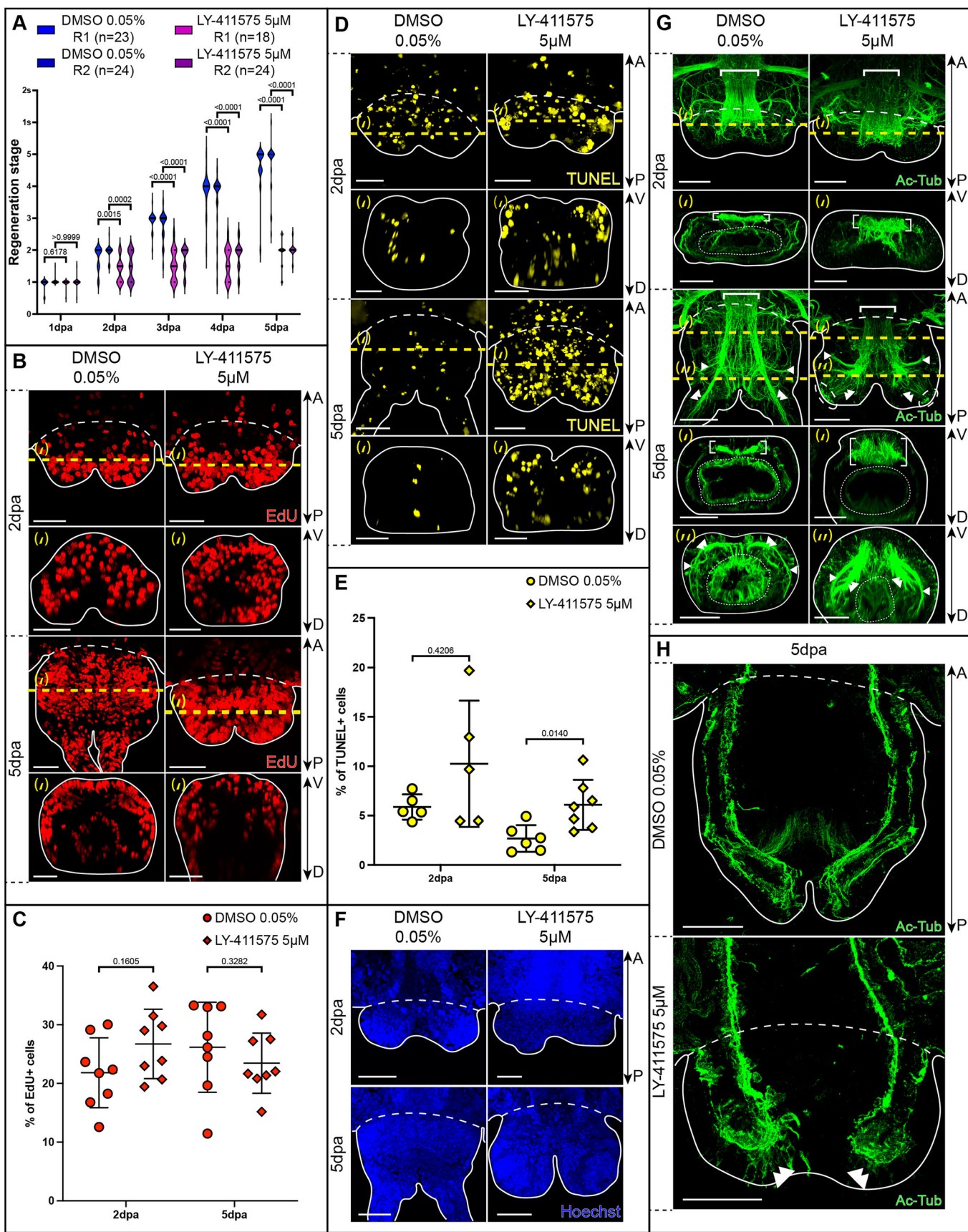

**Figure 2. Morphological and cellular effects of Notch pathway inhibition during posterior regeneration.**

(**A**) Violin plots representing the regeneration stages reached by each animal every day for 5 days upon LY-411575 5 µM treatment in comparison to DMSO 0.05% control. R1 and R2 are independent biological replicates performed for n animals (*n* ranging from 18 to 24). s = segment. (**B**) EdU labelling following a 1 h-pulse. (**C**) Comparison of the proportions of EdU+ cells between conditions at 2 and 5 dpa (*n* = 8 per condition). TUNEL assay (**D**) and comparison of the proportions of TUNEL+ cells (**E**), *n* = 6 per condition); Hoechst nuclei labelling (**F**) and acetylated tubulin (Ac-tub) (**G**) immunolabelling on whole-mount regenerated parts of LY-411575-treated worms and controls at 2 and 5 dpa. (**H**) Acetylated tubulin immunolabelling on longitudinal cross-sections of LY-411575-treated worms and controls at 5 dpa. Data information: (**B–G**) Ventral views are on top and corresponding virtual transverse sections (along the yellow dotted lines (') or ('')) are at the bottom. In all relevant panels, solid white lines delineate the outlines of the samples, white dashed lines correspond to the amputation planes and white dotted lines delineate the gut. White brackets = ventral nerve cord; white arrowheads = circular pygidial nerve; white double arrowheads = nerves of the pygidial cirri; white dashed ellipse = thick acetylated-tubulin+ *foci*. dpa = day(s) post-amputation. Scale bars = 50 µm. Anteroposterior (A/P) and dorsoventral (D/V) axes are represented. For data presented in (**A, C, E**), unpaired Mann–Whitney *U* tests were used for statistical analyses. *P* values as well as mean ± s.d. (for **C, E**) are indicated in the figure and in the Table EV4. Data in (**A, C, E**) are representative of two independent experiments. All images come from representative samples of at least four technical replicates. Source data are available online for this figure.

genes *Engrailed (en)* and *Wnt1* (Prud'homme et al, 2003) (Fig. EV2F,G) were lost in LY-411575-treated worms at 5 dpa (Fig. EV2F'2,G'2). In contrast, the expression patterns of markers of the terminal part of the worm, i.e. the pygidium and its pygidial cirri (*Cdx* and *Dlx*, Fig. EV2H,I) were expanded, suggesting that most of the cells within the regenerating part at 5 dpa have a pygidial identity upon Notch pathway inhibition. Unaltered expression patterns of *FoxA* (Fig. EV2J) together with the presence of an enlarged ring-shaped bundles of pygidial muscles, shown by the expression of *Twist* and *TroponinI* (Fig. EV2K,L) and phalloidin labelling (Fig. EV2M,N) indicated gut and muscle regeneration within the hypertrophied structure. Finally, acetylated tubulin labelling of neurites revealed major nervous system defects in this hypertrophied pygidium (Fig. 2G,H). While the circular pygidial nerve was preserved (white arrowheads), the two nerves of the pygidial cirri (white double arrowheads) were disrupted and multiple aberrant nerve projections extended throughout the 5 dpa regenerated structure and terminated in thick acetylated-tubulin+ *foci* (white dashed ellipse) in its most posterior part (Fig. 2G,H). In addition, at both 2 and 5 dpa, the VNC (white brackets) exhibited a dimmer signal and was noticeably thicker in LY-411575-treated worms.

Taken together, these results indicate that early chemical inhibition of Notch pathway during posterior regeneration in *Platynereis* leads to the formation of an altered terminal structure, thereby hindering regeneration to proceed properly. This structure harbors a dysfunctional growth zone producing few unsegmented tissues and a hypertrophied proliferative pygidium exhibiting increased apoptosis and severe nervous system disruptions.

## Notch signaling controls pygidial neurogenesis during regeneration by regulating *Hes* genes activity

### Transcriptome-wide analysis of Notch pathway inhibition identifies nervous system genes as downstream targets

Given the dramatic nervous system defects induced by Notch pathway inhibition during posterior regeneration, we decided to further explore its impact on gene expression by performing a bulk RNA-seq unbiased approach between LY-411575-treated and control worms at 1 and 2 dpa (Appendix Fig. S3A,B; Table EV1).

Differential gene expression analysis identified 932 differentially expressed genes (DEG) between DMSO control and LY-411575 treated worms at 1 dpa. Among them, 401 are downregulated, 531 are upregulated and 426 are specific to this comparison (Fig. 3A,A';

Dataset EV2). A similar number of DEG were identified between the control and treated worms at 2 dpa (*n* = 1012; 539 are downregulated, 473 are upregulated and 506 are specific to this comparison) (Fig. 3A,A'; Dataset EV3). Gene Ontology (GO) term enrichment analysis indicated that many genes related to stress are upregulated at both 1 and 2 dpa, while genes related to neurogenesis and development are downregulated (Appendix Fig. S4). The thorough manual analysis of the DEG at 1 and 2 dpa (both up and down combined) confirmed and extended this tendency. Genes related to inflammation, immune system, redox signaling and stress, processes well known to be involved in the early steps of regeneration (Bideau et al, 2021; Poss and Tanaka, 2024; Vullien et al, 2025; Vullien et al, 2021) are prominent (Fig. 3B). Importantly, about one fifth (17 and 19% for 1 and 2 dpa respectively) of DEG are related to nervous system. Most of them are upregulated (14 and 13%) and associated more specifically to axon connections or neurite outgrowths regulation, consistent with the acetylated tubulin aberrant phenotype (Fig. 2G,H). Key members of the neurogenic cascade, such as *achaete-scute 1* and *2* (markers of neural progenitors) and *atonal* (a marker of PNS neurogenesis) are upregulated more specifically at 2 dpa (Simionato et al, 2008) (Dataset EV3).

### Notch pathway inhibition triggers excessive neurogenesis and ectopic neuron formation in the hypertrophied regenerated pygidium

Using a set of markers known to be involved in the formation of the larval nervous system (Demilly et al, 2011; Denes et al, 2007; Kerner et al, 2009; Simionato et al, 2008), we assessed the effects of Notch pathway inhibition on the developmental neurogenic cascade during pygidium regeneration (Fig. 4).

First, Notch pathway inhibition altered the expression patterns of several neural progenitor markers (*Achaete-scute 1* and *2 (Acs1* and *2), Neurogenin (Ngn)*). At 2 dpa, *Acs1* and *2* were markedly upregulated in the blastema of treated worms (Fig. 4A2,B2) whereas *Ngn* expression remained unchanged compared to the control (Fig. 4C2). At 5 dpa, these three markers were broadly expressed in the hypertrophied pygidium in LY-411575-treated worms (Fig. 4A'2,B'2,C'2) in stark contrast to their restricted expression in controls (in particular for *Acs1* and *Ngn*; Fig. 4A'1,C'1). Notably, *Acs1* and *2* were strongly downregulated in the regenerating neuroectoderm in treated worms (Fig. 4A'2,B'2), likely due to the severe reduction in tissue growth. We then determined the expression patterns of two markers of differentiating neurons: *Elav*, found in all differentiating neurons (Fig. 4D) and

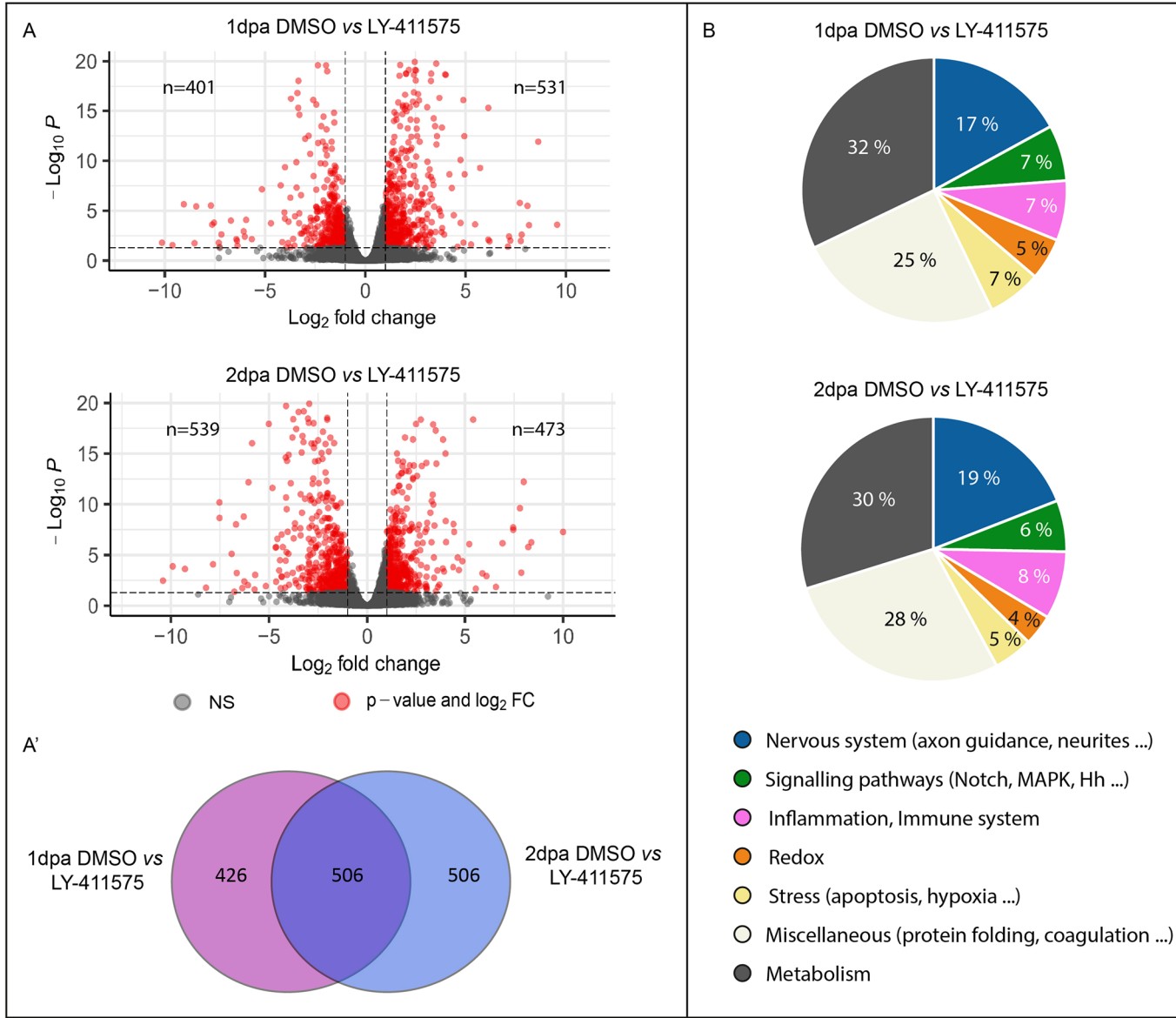

**Figure 3. Impacts of Notch pathway inhibition on transcriptome-wide gene expression during regeneration.**

(**A**) Volcano plots comparing LY-411575-treated regenerated parts *vs* controls at 1 (top) and 2 dpa (bottom). Each dot represents a transcript. Differentially expressed genes (DEGs) are depicted in red, for which FDR < 0.05 and logFC < −1 or logFC >1. *P* values were calculated using the likelihood ratio test and adjusted for multiple tests using the Benjamini–Hochberg FDR correction. (**A'**) Venn diagram representing the DEGs specific or common among 1 and 2 dpa conditions. (**B**) Pie charts depicting the functional classifications of the DEGs for 1 and 2 dpa conditions. dpa = day(s) post-amputation. Source data are available online. Three independent replicates were performed.

*Collier*, exclusively expressed in the VNC (Fig. 4E). At 2 dpa, the expression patterns of both markers were similar between LY-411575-treated and control worms (Fig. 4D1,D2,E1,E2). However, at 5 dpa, *Elav* expression was dramatically upregulated and had expanded throughout the whole mesodermal compartment of the hypertrophied pygidium in treated worms, in contrast to control animals where its expression remained restricted to a few cells at the base of the pygidium and in its cirri (Fig. 4D'1,D'2). Additionally, *Elav* expression in the anterior part of the regenerated region, including in the VNC, was reduced in treated worms. Although *Collier* expression in the VNC was maintained (Fig. 4E'2),

an ectopic expression domain emerged labelling differentiating neurons positioned transversely within the pygidium (Fig. 4E'2). *Synaptotagmin (Syt)*, a broad marker of differentiated neurons, exhibited a massively expanded expression in LY-411575-treated worms compared to controls at 5 dpa (Fig. 4F'1,F'2). Moreover, while the cholinergic marker *VAChT* and the glutamatergic marker *VGluT* were unaffected at 2 dpa (Fig. 4G1,G2,H1,H2), *VGluT* expression was distinctly extended in the hypertrophied pygidium at 5 dpa (Fig. 4H'2).

To further examine the pygidial defects in LY-411575-treated worms at 5 dpa, we combined immunolabelling for acetylated

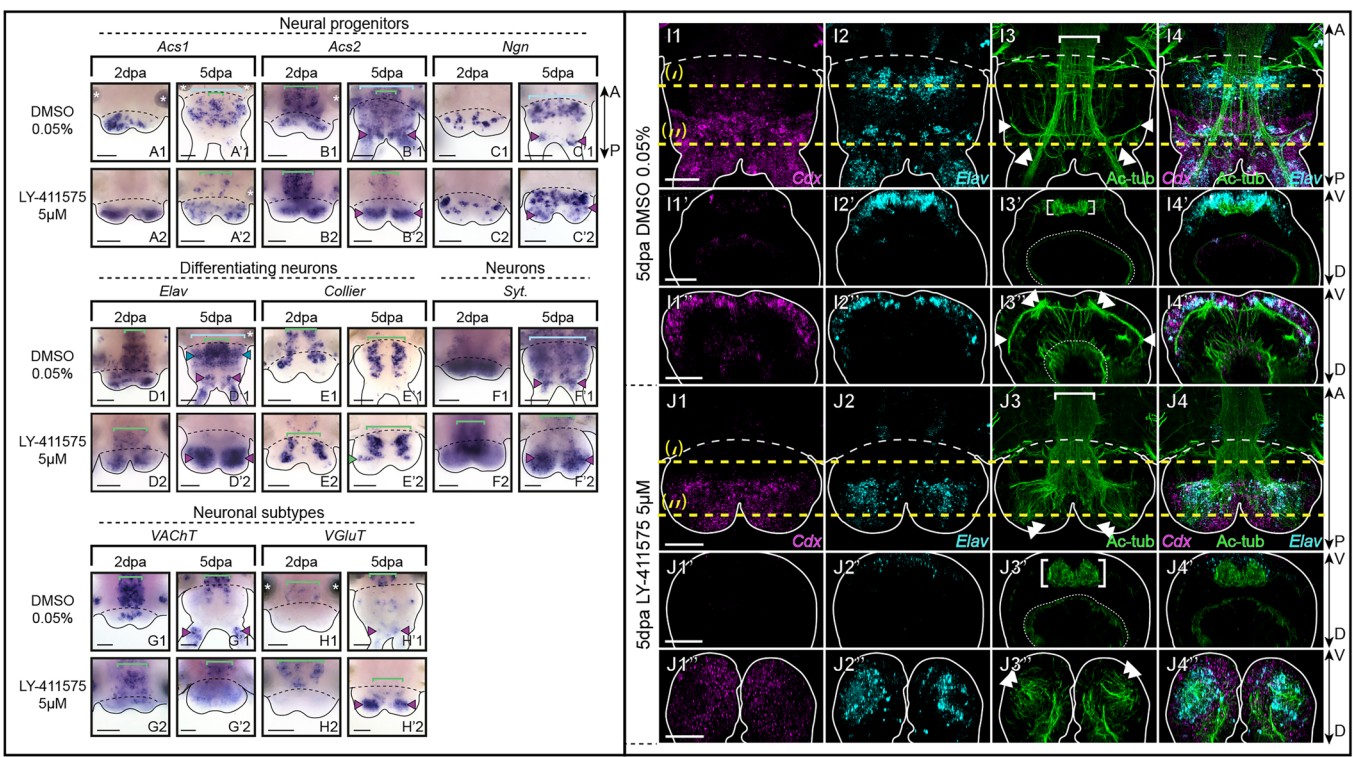

**Figure 4. Notch signaling pathway inhibition during posterior regeneration leads to major neural defects in the pygidium.**

(A–H) Whole-mount in situ hybridizations for markers of neural progenitors (**A–C**), differentiating neurons (**D, E**), differentiated neurons (**F**) and neuronal subtypes (**G, H**) for LY-411575 treated worms and controls at 2 and 5 dpa, ventral views. Green brackets = ventral nerve cord (VNC); light blue brackets = neuroectoderm; purple arrowheads = neurons of the pygidium and cirri; green arrowheads = ectopic VNC neurons; white asterisks = non-specific staining from glands. Solid black lines delineate the outlines of the samples, black dashed lines correspond to the amputation planes. (**I, J**) Hybridization chain reaction (HCR) for *Cdx* (purple, 1 and 4) and *Elav* (cyan, 2 and 4) coupled with immunolabelling for acetylated tubulin (green, 3 and 4) for LY-411575 treated worms and controls at 5 dpa. Ventral views are on top and corresponding virtual transverse sections along the yellow dotted lines (' and ", respectively) are at the bottom. Solid white lines delineate the outlines of the samples, white dashed lines correspond to the amputation planes. White brackets = VNC; white arrowheads = circular pygidial nerve; white double arrowheads = cirri nerves. dpa = day(s) post-amputation. Scale bars = 50 µm. Anteroposterior (A/P) and dorsoventral (D/V) axes are represented. All images come from representative samples of two biological replicates. Source data are available online for this figure.

tubulin with hybridization chain reaction (HCR) for the pygidial marker *Cdx* and the marker of differentiating neurons *Elav* (Fig. 4I,J). As expected in control worms, *Cdx* was expressed in the whole ectoderm of the pygidium (Fig. 4I1,I1") and was absent in the anterior tissues produced by the GZ (Fig. 4I1'). *Elav* was expressed both in the VNC (Fig. 4I2,I2') and in the ectoderm of the pygidium, as well as in the pygidial cirri (Fig. 4I2"). As described earlier, acetylated tubulin labelling delineated the VNC (Fig. 4I3,I3'), the pygidial nerve (Fig. 4I3,I3"), the nerve net around the gut (Fig. 4I3',I3") and the nerve extensions in the pygidial cirri (Fig. 4I3). Virtual transverse sections revealed that the VNC was overlaid by *Elav*+ cells in the anterior regenerating region (Fig. 4I4,I4'), while *Elav*+ and *Cdx*+ cells, in the pygidial ectoderm, enclosed the pygidial nerve (Fig. 4I4"). In LY-411575-treated worms (Fig. 4J), *Cdx* signal was abnormally found in the whole pygidium, including internal tissues (Fig. 4J1,J1") while *Elav* was only maintained in few cells of the thickened VNC (Fig. 4J2,J2') but also strongly expressed in many internal cells within the hypertrophied pygidium (Fig. 4J2"). Those *Cdx*+ and/or *Elav*+ cells were encompassing the abnormal nerve projections observed in the modified pygidium (Fig. 4J4,J4").

While this neural phenotype might be interpreted as indirectly resulting from the slight disorganization of the regenerated pygidium, we consider much more likely that Notch signaling inhibition is the primary cause of these major neural defects. Altogether, our results suggest that Notch signaling acts as a key regulator of pygidial neurogenesis by controlling neural progenitor specification. When Notch is inhibited, neural progenitor genes are misregulated leading to the overproduction of neurons, including glutamatergic subtypes, and ultimately resulting in a hypertrophied and disorganized pygidium.

### Notch pathway inhibition affects the expression of the ligand Delta and several Hes target genes during pygidial neurogenesis

To elucidate how the Notch pathway components are involved in pygidial neurogenesis during regeneration, we assessed their expressions upon LY-411575 treatment (Fig. 5; Dataset EV4). While *Notch* receptor, its potential ligands *DSL1* and *3* as well as *Presenilin* did not appear affected by the treatment (Fig. 5A,C–E), the expression of the ligand *Delta* extended throughout the hypertrophied pygidium at 2 dpa (Fig. 5B2) and displayed an intense ectopic expression in two lateral patches of ectodermal cells at 5 dpa (Fig. 5B'2). Among the 7 *Hes*-related genes found in

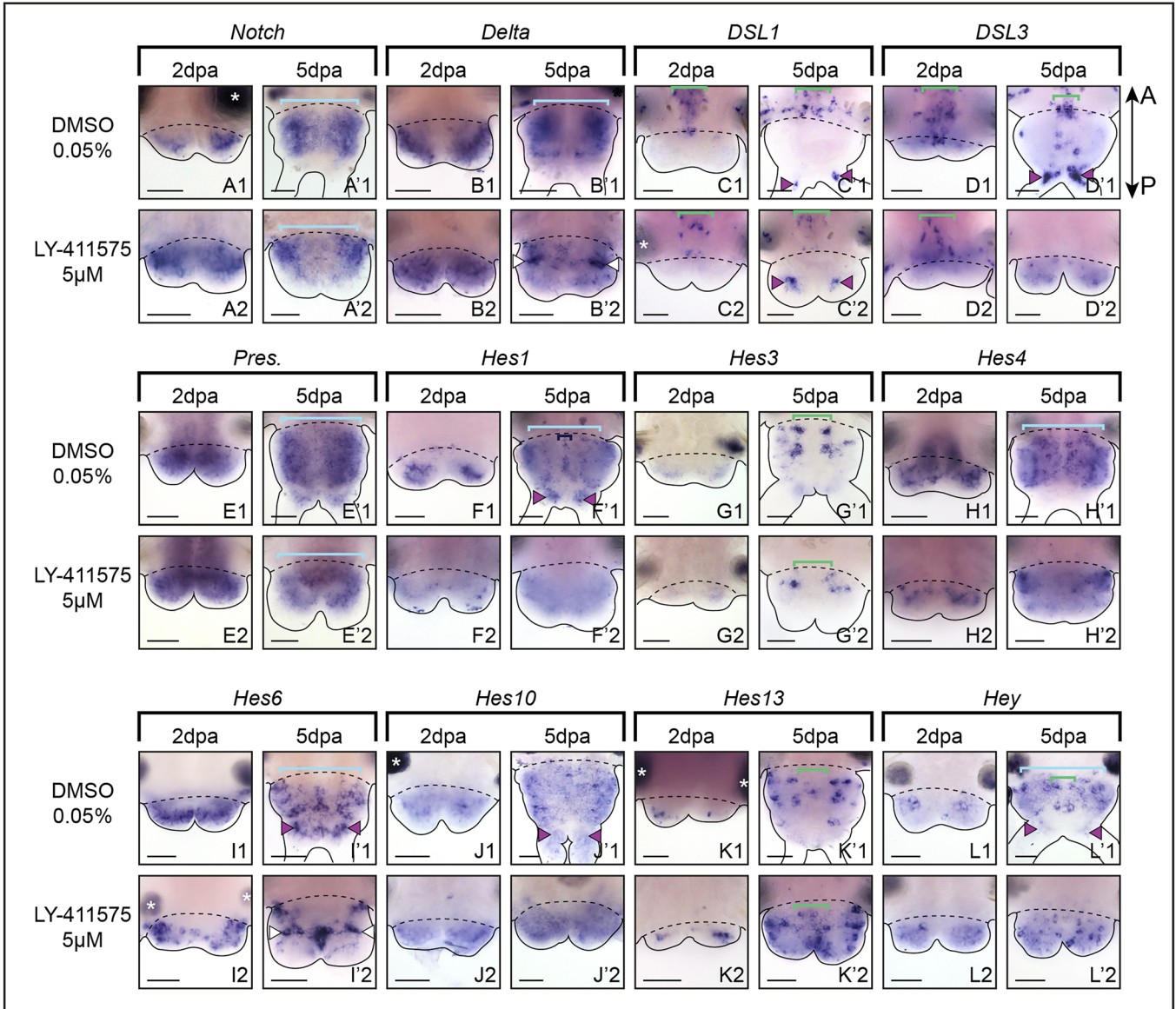

**Figure 5. Notch pathway inhibition alters the expression patterns of several core members of the pathway and its putative targets in the nervous system during posterior regeneration.**

Whole-mount in situ hybridizations for core components of Notch (**A–E**) as well as putative targets of the pathway (**F–L**) for LY-411575 treated worms and controls at 2 and 5 dpa. Ventral views. White arrowheads point to lateral patches of ectodermal cells harboring intense expression of *Delta* and *Hes6*. Green brackets = ventral nerve cord; light blue brackets = neuroectoderm; purple arrowheads = neurons of the pygidium and cirri; dark blue brackets = midline; white asterisks = non-specific staining from glands. Solid black lines delineate the outlines of the samples, black dashed lines correspond to the amputation planes. dpa = day(s) post-amputation. Scale bars = 50 μm. Anteroposterior (A/P) axis is represented. All images come from representative samples of at least two biological replicates. Source data are available online for this figure.

neurogenic structures, 4 of them showed a modified expression pattern (*Hes1, 4, 6, 13*; Fig. 5F,H,I,K), while *Hes3, 10* and *Hey* were not altered in LY-411575-treated worms (Fig. 5G,J,L). *Hes1*+ and *Hes4*+ territories were markedly reduced at 2 and 5 dpa (Fig. 5F2,F'2,H2,H'2), supporting the downregulation of these genes found in the comparative bulk RNA-seq data at early stages (Dataset EV4). *Hes6* expression at 5 dpa was also extremely altered in comparison to the DMSO control (Fig. 5I2,I'2), with much less *Hes6*+ cells in the neuroectoderm and a shift of expression in two

lateral ectodermal patches (Fig. 5I'2), as observed for *Delta* (Fig. 5B'2), and in a central ectodermal region of the pygidium and the anus. Finally, the overexpression of *Hes13* already revealed by the RNA-seq data is supported by its extended expression at both 2 and 5 dpa (Fig. 5K2,K'2; Dataset EV4).

As expected, Notch pathway inhibition also leads to significant alterations in the expression of Notch pathway components and putative *Hes* target genes expressed in non-neurogenic territories, particularly at 5 dpa (Fig. EV3; Dataset EV4). A reduced number of

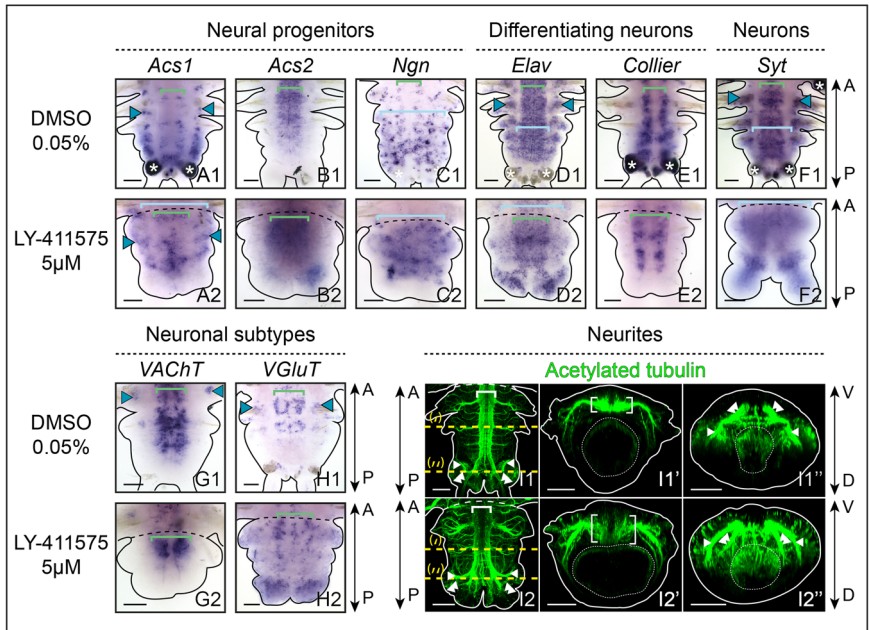

**Figure 6. Notch pathway inhibition during post-regeneration posterior elongation leads to major neural defects in the ventral nerve cord.**

(A–H) Whole-mount in situ hybridizations for markers of neural progenitors (A–C), differentiating neurons (D, E), differentiated neurons (F) and neuronal subtypes (G, H) for LY-411575 treated worms from 3 dpa to 10 dpa and DMSO controls. Ventral views. Green brackets = ventral nerve cord; light blue brackets = neuroectoderm; blue arrowheads = PNS; white asterisks = non-specific staining from glands. Solid black lines delineate the outlines of the samples; black dashed lines correspond to the amputation planes. (I) Acetylated tubulin immunolabelling on whole-mount regenerated parts of LY-411575-treated from 3 dpa to 10 dpa worms and controls at 10 dpa. Ventral views are on the left and corresponding virtual transverse sections along the yellow dotted lines (' and '', respectively) are on the right. Solid white lines delineate the outlines of the samples; white dashed lines correspond to the amputation planes and white dotted lines delineate the gut. White brackets = ventral nerve cord; white arrowheads = circular pygidial nerve; white double arrowheads = cirri nerves. Scale bars = 50 μm. Anteroposterior (A/P) and dorsoventral (D/V) axes are represented. All images come from representative samples of at least two biological replicates. Source data are available online for this figure.

*Hes5+* and *Hes8+* cells was observed in LY-411575-treated worms (Fig. EV3D'2,E'2) while *Nrarp*, *Hes2* and *Hes8* were no longer expressed in the GZ (Fig. EV3B'2,C'2,E'2).

Altogether, we found that the Notch pathway, potentially through the action of its ligand *Delta* and several *Hes* genes, regulates the specification of neural progenitors and their differentiation into neurons in the regenerated pygidium leading to its hypertrophy upon the inhibition of the pathway. Since Notch is crucial for regenerating a functional growth zone, its inhibition impairs new tissue production and leads to an improperly formed VNC.

## Notch signaling controls central nervous system neurogenesis during post-regenerative posterior elongation in *Platynereis*

We next investigated the effects of Notch pathway inhibition on the VNC reformation during post-regenerative posterior elongation. To circumvent the fact that Notch pathway inhibition induces a dysfunctional regenerated GZ—characterized by impaired tissue production—we initiated LY-411575 treatments after 3 dpa, once the GZ has already reformed (Planques et al, 2019). This staggered treatment regimen (Fig. EV4A) enables the formation of elongated tissues in which Notch-dependent effects on the CNS neurogenesis can be studied (Fig. EV4B). Also, posterior elongation uniquely recapitulates the temporal progression of CNS and PNS

neurogenesis in a postero-anterior manner: early neurogenic events occur in newly produced, growth zone-derived tissues, while later stages are visible more anteriorly.

Notch pathway inhibition performed during posterior elongation (from 3 to 10 dpa) led to dramatic CNS defects. Neural progenitors markers were disrupted: *Acs2* expression was lost from superficial cells of the VNC (Fig. 6B2) while *Acs1* and *Ngn* expressions were broadly expanded throughout the neuroectoderm, exceeding their normal localization in neurogenic columns (Fig. 6A2,C2). Similarly, *Elav* and *Syt* showed expanded neuronal domains (fully differentiated or not; Fig. 6D2,F2). However, the patterns of *Collier*, *VAChT* and *VGluT* remained relatively unaffected (Fig. 6E,G,H). Acetylated tubulin labelling revealed abnormal nerve projections in the modified pygidium (Fig. 6I2,I2''), as previously observed at early stages. The VNC exhibited reduced neurite density, an abnormal U-shape, and was mispositioned deeper within the ventral tissues (Fig. 6I2,I2'). Additionally, nerve projections within the neuroectoderm were highly disorganized (Fig. 6I2,I2'), coinciding with increased apoptosis (Appendix Fig. S5).

To further assess how Notch pathway inhibition alters the three-dimensional architecture of the neuroectoderm during CNS neurogenesis, we performed HCR for key neurogenic markers (*Ngn, Elav, Syt*) known to specify larval trunk neuroectoderm cells layers (Figs. 7 and EV5) (Demilly et al, 2013). Indeed, these genes recapitulate neurogenic events throughout the stratification of the

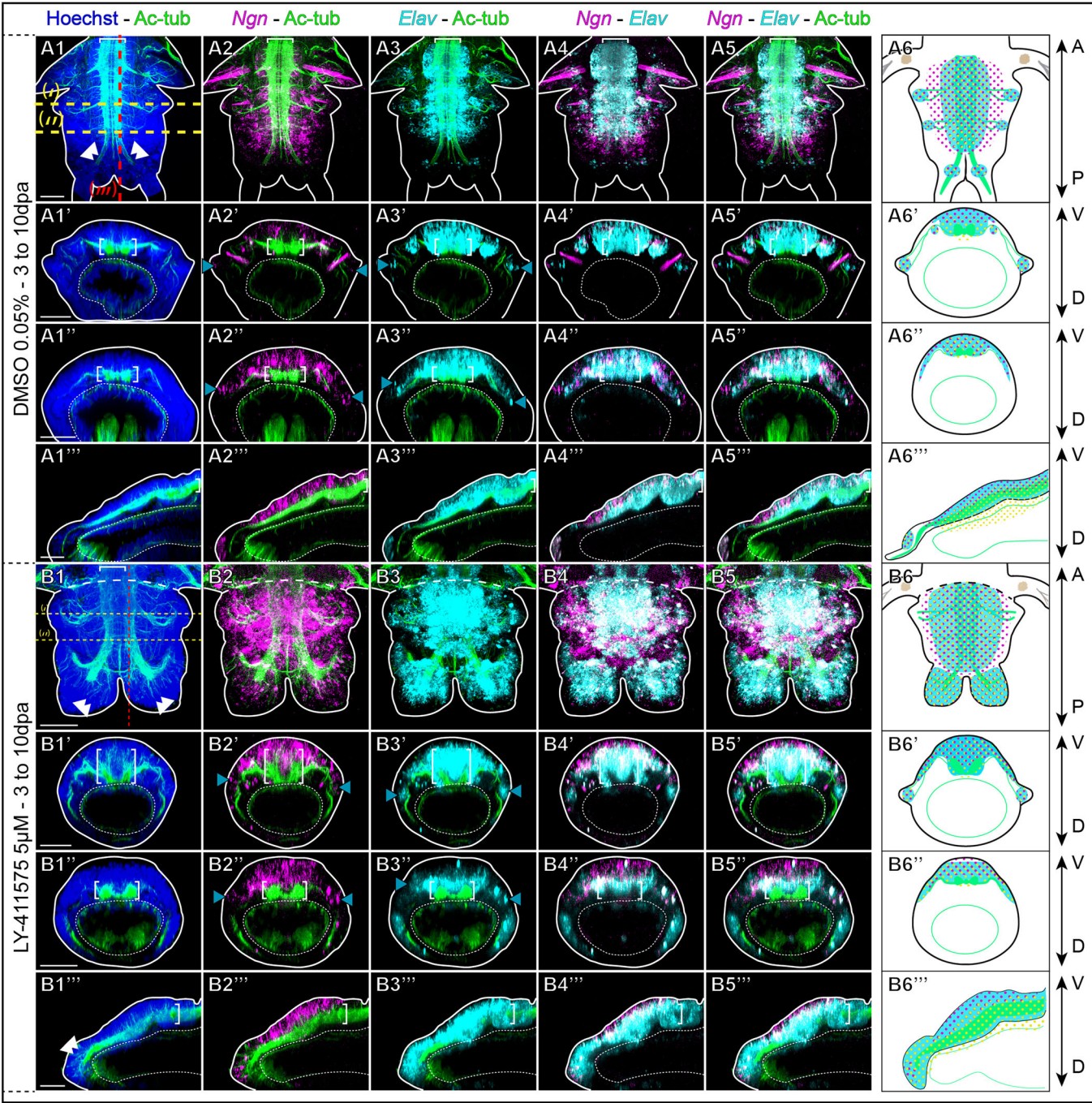

**Figure 7. Notch pathway inhibition during post-regeneration posterior elongation leads to a disorganized thicker ventral nerve cord and to the enlargement of the neuroectodermic territories.**

Hybridization chain reactions (HCR) for *Elav* (cyan) and *Ngn* (magenta) coupled with immunolabelling for acetylated tubulin (green) and nuclei staining with Hoechst (blue) for controls at 10 dpa (top, **A**) and LY-411575 treated regenerated parts from 3 dpa to 10 dpa (bottom, **B**). Ventral views are on top for each condition and corresponding virtual transverse sections (along (') and ('') in yellow) and sagittal section ((''') in red) are at the bottom. On the right, schematic drawings depict the expression patterns of *Elav* (cyan), *Ngn* (magenta dots) and *Syt* (yellow dots – from Fig. EV5) as well as their arrangement around the main structures of the central nervous system and peripheral nervous system (green). Solid white lines delineate the outlines of the sample, and white dotted lines delineate the gut. White brackets = ventral nerve cord; white double arrowheads = cirri nerves; blue arrowheads = PNS. dpa = day(s) post-amputation. Scale bars = 50 μm. Anteroposterior (A/P) and dorsoventral (D/V) axes are represented. All images come from representative samples of two biological replicates. Source data are available online for this figure.

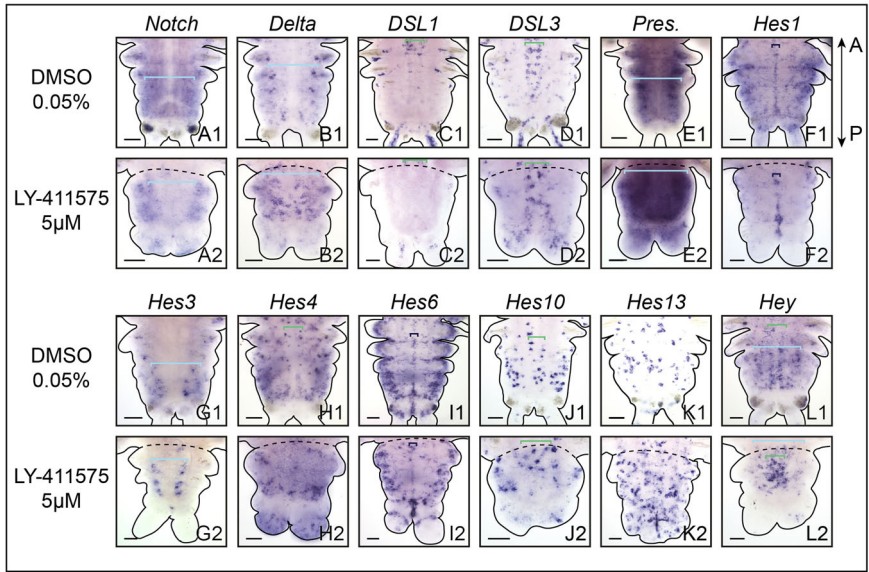

**Figure 8. Notch pathway inhibition during post-regeneration posterior elongation alters the expression patterns of several core members of the pathway and its putative targets in the nervous system.**

Whole-mount in situ hybridizations for core components of Notch (**A–E**) as well as putative targets of the pathway (**F–L**) for LY-411575-treated worms from 3 dpa to 10 dpa and controls at 10 dpa. Ventral views. Green brackets = ventral nerve cord; light blue brackets = neuroectoderm; purple arrowheads = cirri; dark blue brackets = midline; white asterisks = non-specific staining from glands. dpa = day(s) post-amputation. Solid black lines delineate the outlines of the samples, black dashed lines correspond to the amputation planes. Scale bars = 50 µm. Anteroposterior (A/P) axis is represented. All images come from representative samples of at least two biological replicates. Source data are available online for this figure.

larval neuroectoderm. On top of the neuroectoderm is a highly proliferative *Ngn*+ expressing layer of neural progenitors, next is a reduced proliferative *Elav*+ expressing layer of differentiating progenitors, with below a post-mitotic *Elav*+ and *Syt*+ expressing layer of maturing neurons and finally a VNC composed of mature *Syt*+ neurons from which acetylated tubulin-labelled axons are projecting (Demilly et al, 2013). During post-regenerative posterior elongation, neurogenesis broadly follows the same sequence of events (Figs. 7A and EV5A). In control worms, the fully differentiated nuclei-dense neuroectoderm (tissues far from the GZ, Figs. 7 and EV5 - Transverse sections (')), had a distinct laminar organization. The superficial cell layers of the CNS neuroectoderm expressed *Ngn* and *Elav* (Fig. 7A2'–A6'), the intermediate layers co-expressed *Ngn, Elav* and *Syt* (Figs. 7A2'–A6' and EV5A2'–A5'), while the deeper ones expressed only *Elav* and *Syt* (Figs. 7A6' and EV5A2'–A5'). Finally, a *Syt*+ cell layer corresponded to the VNC (white brackets) from which neurites were projecting (Figs. 7A6' and EV5A2'–A5'). Lateral nerves innervated the ganglions of the PNS, which are *Ngn* +, *Elav*+ and *Syt* + (Figs. 7A1'–A6' and EV5A1'-A5'). During early neurogenesis (tissues close to the GZ, Figs. 7 and EV5 - Transverse sections ('')), superficial thin layers of the CNS neuroectoderm were composed of *Ngn*+ and *Elav*+ cells (Fig. 7A2''–A6''), while deeper cells were co-expressing *Ngn, Elav* and *Syt* (Figs. 7A2''–A6'' and EV5A2''–A5''). As for late neurogenesis, the VNC was positioned below the whole CNS neuroectoderm, while lateral nerve extensions supported the PNS neuroectoderm anlagen composed of *Ngn*+ and *Elav*+ cells (Fig. 7A1''–A6''). Sagittal sections (Figs. 7 and EV5''') illustrated that *Ngn* (Figs. 7A2''',A4'''–A6''') is the earliest marker to be expressed in

superficial neuroectodermal cells, with its expression diminishing as differentiation proceeds. Conversely, *Elav* and *Syt* were expressed subsequently and maintained throughout CNS maturation (Figs. 7A3'''–A6''' and EV5A2'''–A5''').

In the context of Notch pathway inhibition, HCR and acetylated-tubulin co-labelling confirmed the drastic alteration of the regenerated neural structures (Figs. 7B and EV5B). In the fully differentiated neuroectoderm of LY-411575-treated worms (tissues far from the GZ), the abnormally U-shaped VNC was overlaid by a markedly thicker nuclei-dense CNS neuroectoderm (Figs. 7B1' and EV5B1'), in which most of the cells were *Ngn* +, *Elav*+ and *Syt* + (Fig. 7B6'). This territory extended on superficial cells until joining the PNS ganglions (Fig. 7B6'). At the level of the VNC, the deepest cell layers of this laterally and thickly enlarged structure contained only *Elav*+ and *Syt*+ cells (Fig. 7B6'). During early neurogenesis (tissues close to the GZ, Transverse section ('')), the two cords of the VNC were slightly apart and the neuroectoderm was already slightly thickened (Figs. 7B1'' and EV5B1''). This CNS neuroectoderm was composed of cell layers that were, from top to bottom, *Ngn* +, then *Ngn* +, *Elav*+ and *Syt*+ and finally *Elav*+ and *Syt* + (Fig. 7B6''). All of these three markers appeared to be expressed almost concomitantly in very recently-produced cells from the GZ and their expression was broadly maintained throughout CNS postero-anterior differentiation (Fig. 7B6'''). Hence, Notch pathway inhibition appears to disturb the dynamics of the neurogenic cascade at the neural progenitor determination step, leading to a thickened CNS neuroectoderm and an abnormally-shaped VNC with neurite defects.

We then dissected how Notch pathway components regulate CNS neurogenesis (Fig. 8) and showed that while *Notch* receptor,

the potential ligands *DSL1* and *3* as well as *Presenilin* did not appear much affected by the treatment (Fig. 8A,C–E), the ligand *Delta* expression was enhanced in the CNS neuroectoderm at 10 dpa (Fig. 8B2). Upon Notch pathway inhibition, 3 of the 7 *Hes*-related genes expressed in neurogenic structures did not appear affected (*Hes3, 6* and *10*, Fig. 8G,I,J) while *Hes1* and *Hes4* were downregulated (Fig. 8F2,H2). *Hes1* expression was maintained in the midline (Fig. 8F2). In contrast, we observed that *Hes13* and *Hey* expression domains were extended in LY-411525 treated worms (Fig. 8K2,L2). Many more *Hes13*+ cells were found in the whole ventral surface of the regenerated structure, likely in both the CNS and PNS, in a salt and pepper fashion, in comparison to controls (Fig. 8K2). *Hey* expression extended to the whole CNS neuroectoderm in a disorganized manner when Notch pathway was inhibited (Fig. 8L2).

Additional data on Notch pathway members altered expressions in LY-411575 treated worms show that Notch pathway is not restricted to neurogenic functions during post-regenerative posterior elongation. Indeed, *Delta*, *Nrarp* and *Hes2* expressions in the GZ were lost in LY-411575 treated worms (Fig. 8B2; Appendix Fig. S6A2,B2). This supports the idea that the GZ may not function perfectly in the absence of Notch pathway signalization, potentially explaining the smaller size of the LY-411575-treated regenerated parts. While this slightly reduced tissue production by the GZ could lead to some of the neural defects observed upon LY-411575 treatment, it appears very unlikely that it could be the main cause of all the neurogenic phenotypes. In addition, Notch determines the identity of the two main cell types composing the chaetal sacs, the follicle cells and chaetoblasts (Gazave et al, 2017; Zakrzewski, 2011). While *Nrarp* and *Hes2* expressions in the follicle cells appeared reduced or lost upon Notch pathway inhibition (Appendix Fig. S6A2,B2), *Hes12* expression in the chaetoblast was extended, and *Hes12*+ cells were even present in the pygidium (Appendix Fig. S6C2). The main other tissues and structures (growth zone, progenitors, appendages and muscles), albeit smaller appeared to be normal, with the exception of the pygidium which was enlarged as observed during the regeneration experiments (Appendix Fig. S6D–H).

In conclusion, we showed with high confidence that Notch pathway regulates neurogenesis during both pygidial regeneration and CNS patterning during posterior elongation.

# Discussion

Notch signaling is a key cell-cell communication pathway that orchestrates diverse cellular processes in both embryonic and post-embryonic developmental contexts across vertebrates and ecdysozoans. Its wide range of functions is enabled by the modular interactions among receptors, ligands, and a variety of target genes (Andersson et al, 2011; Bray, 1998; Henrique and Schweisguth, 2019). However, the role of Notch signaling in spiralians remains relatively unexplored, which is limiting our understanding of its ancestral functions at the bilaterian scale. Here, we uncovered the multiple functions of Notch signaling during posterior regeneration and elongation in the annelid *Platynereis*. This spiralian model possesses a complete set of core Notch components—including the receptor *Notch*, ligands *Delta* and *Jagged*, the regulator *Nrarp* and an array of *Hes/Hey* effectors (Gazave et al, 2014; Iso et al, 2003).

Our study shows that Notch pathway in *Platynereis* is remarkably modular, participating in several distinct processes. We found that Notch signaling is essential for the proper regeneration of the growth zone stem cells, for the formation of bristles during posterior elongation (as previously described during larval development (Gazave et al, 2017)) and for the regulation of neurogenesis during both post-embryonic processes likely *via* the transcriptomic modulation of up to eight *Hes/Hey* genes.

In particular, our comprehensive molecular analysis of the neurogenic cascade involved in pygidial regeneration and ventral nerve cord formation through posterior elongation, revealed that impairing Notch signaling leads to excessive neurogenesis. This indicates that Notch signaling plays a pivotal role in maintaining the balance between neural progenitor specification and the production of differentiated neurons. Surprisingly, previous research on Notch signaling did not find supporting evidence for such effects during early larval neurogenesis (Gazave et al, 2017).

In vertebrates and *Drosophila*, Notch signaling is well known for its function in neuronal cell fate specification, where it preserves a pool of progenitors while directing differential cell fates, such as neurons *versus* glial cells (Chouly and Bally-Cuif, 2024; Pierfelice et al, 2011). In *Platynereis*, it remains unclear whether the excess neurons produced upon Notch signaling inhibition comes at the expense of other cell types (e.g. glial cells), due in part to the current lack of a specific glial marker. Nevertheless, our data support a conserved and ancestral role for Notch signaling in regulating neuronal balance across bilaterians. A recent study in the planarian *Schmidtea* revealed that Notch signaling is also regulating glial specification during regeneration *via* the interactions between mature neurons and non-neural progenitors, suggesting a conserved role of Notch signaling in glial development specification rather than on neuronal balance (Scimone et al, 2025). Discriminating between these two hypotheses will require more research in additional spiralian models.

Our study also illustrates a potential role of Notch signaling in regulating axon guidance and neurite outgrowth. *Platynereis* worms have a complex VNC and pygidial neural network composed of highly organized neuropils that rely on precise axon guidance mechanisms (Demilly et al, 2013). In the presence of the γ-secretase inhibitor, we observed aberrant axonal projections during both pygidial nervous system regeneration and VNC formation in posterior elongated tissues, leading to a scattered and enlarged VNC. We found a concomitant increase of cell death by apoptosis, possibly due to the failure of misrouted axons to establish proper target connections (Vanderhaeghen and Cheng, 2010). Notch has been shown to regulate axon patterning and neurite outgrowth in both vertebrates (Aujla et al, 2011; Shi et al, 2011) and *Drosophila* (Kannan et al, 2018; Kuzina et al, 2011; Zhang et al, 2023). Although we cannot exclude the possibility that other γ-secretase substrates may act on these processes, our findings suggest that Notch-mediated axon guidance might represent an ancestral function in protostomes and even bilaterians.

Our work highlights the multifaceted and conserved roles of Notch signaling in bilaterian neurogenesis, ranging from the upstream neural progenitors' specification to potentially the ultimate axon guidance step allowing fine neural circuitry. These findings underscore the need for further studies in diverse model organisms to fully elucidate its evolutionary and developmental significance.

## Limitations of the study

As inducible knockout techniques have not yet been developed for regenerating *Platynereis* worms, our study relies on the use of γ-secretase inhibitors (GSI) to block Notch signaling pathway. Although GSI are broadly used in multiple model organisms to this end (Dirian et al, 2014; Foster et al, 2025; Haillot et al, 2025; Najle et al, 2023; Narayanaswamy et al, 2025; Romero-Carvajal et al, 2015; Wu et al, 2016; Zhao et al, 2022), these inhibitors can hinder the cleavage of other membrane proteins besides Notch, including the *Netrin* receptor DCC which is involved in neurite outgrowth and axon guidance (Guner and Lichtenthaler, 2020). To test the hypothesis that DCC /Netrin pathway inhibition could lead to the phenotype we observed, we identified its downstream genes in our differential transcriptome and found that none were affected upon GSI treatment (Table EV2). Next, we determined that chemical inhibition of Src family kinases (SFKs), a gene family known as effectors of Netrin/DCC pathway, did not produce the same phenotype as the one obtained using GSI (see Appendix Fig. S7). Both these elements support the fact that this key other target of GSI is not responsible of the observed neural phenotype. However, such experimental approach cannot rule out the possibility that, in addition to Notch, other pathways might also be altered by treatments with GSI and could partly mediate the observed neural phenotype. Finally, the precise roles of *Delta* and *Hes* genes in *Platynereis* post-embryonic neurogenesis remains to be determined.

# Methods

### Reagents and tools table

| Reagent/resource | Reference or source | Identifier or catalog number |
| --- | --- | --- |
| **Experimental models** | | |
| *Platynereis dumerilii* | In-house culture | -- |
| **Recombinant DNA** | | |
| **Antibodies** | | |
| Mouse anti-acetylated tubulin Monoclonal antibody | Merck | T7451-25UL |
| Anti-digoxigenin- AP conjugated antibody | Merck | 11093274910 |
| Fluorescent secondary antibodies anti-mouse IgG Alexa Fluor | Cell Signalling | 4408S (488) 4413S (555) |
| **Oligonucleotides and other sequence-based reagents** | | |
| HCR probes | This study | Dataset EV5 |
| HCR™ Amplifier: B1-647 | Molecular Instruments, Inc. | – |
| HCR™ Amplifier: B3-594 | Molecular Instruments, Inc. | – |
| Q5 High-fidelity DNA polymerase | New England Biolabs | M0491S |
| **Chemicals, enzymes and other reagents** | | |
| Click-it EdU Imaging Kit | ThermoFisher | C10338 (488) C10337 (555) |
| Click-iT TUNEL kit 647 | ThermoFisher | C10247 |

| Reagent/resource | Reference or source | Identifier or catalog number |
| --- | --- | --- |
| Phalloidin-Alexa 555 | Molecular Probes | A34055 |
| Hoechst 33342 | Molecular Probes | C10338 |
| OCT embedding medium | CellPath | KMA-0100-00A |
| NBT | Roche | 11383213001 |
| BCIP | Roche | 11383221001 |
| Proteinase K | Ambion | AM2548 |
| DAPT | Med Chem Express | HY-13027 |
| LY-411575 | Med Chem Express | HY-50752 |
| RO-4929097 | Med Chem Express | HY-11102 |
| PP2 | Med Chem Express | HY-13805 |
| RNAqueous total RNA Isolation kit | Ambion | AM1912 |
| RNA 6000 Nano kit for Bioanalyzer | Agilent | 5067-1511 |
| DIG | Roche | 11277073910 |
| Protector RNAse inhibitor | Roche | 03335399001 |
| T7 RNA polymerase | Roche | 10881767001 |
| Sp6 RNA polymerase | Roche | 10810274001 |
| Agarose D5 | Euromedex | D5-C |
| NucleoSpin RNA | Macherey-Nagel | 740955.50 |
| Sheep serum (for Ab) | Merck | S3772-5ML |
| Sheep serum (for ISH) | Merck | S22-100ML |
| Denhardt's Solution 50x | Merck | D2532-5ML |
| Sucrose | Sigma-Aldrich | 50389-500 g |
| DMSO | Euromedex | UD8050-B |
| MgCl2 hexahydrate | Supelco | 1.05833.0250 |
| Paraformaldehyde | Sigma-Aldrich | 158127-500 g |
| Tween20 | Sigma-Aldrich | P7949-500ml |
| Dextran Sulfate sodium salt | Sigma-Aldrich | 42867-5 G |
| Formamide | Sigma-Aldrich | 252549-500 ml |
| DABCO | Sigma-Aldrich | 027802-100 g |
| DIG probes | This study | Table EV3 |
| **Software** | | |
| Imaris 9.5.0 | Oxford Instruments http://www.bitplane.com/imaris/imaris | RRID:SCR_007370 |
| FIJI | https://doi.org/10.1038/nmeth.2019 | |
| ProbeMaker | https://doi.org/10.1002/jez.b.23100 | |
| GraphPad Prism 9 | www.graphpad.com | |
| FastQC v0.11.8 | https://www.bioinformatics.babraham.ac.uk/projects/fastqc/ | |
| Fastp v0.23.2 | https://doi.org/10.1093/bioinformatics/bty560 | |
| Kallisto v0.48.0 | https://doi.org/10.1038/nbt.3519 | |

| Reagent/resource | Reference or source | Identifier or catalog number |
|---|---|---|
| Trinity v2.13.2 | https://doi.org/10.1038/nprot.2013.084 | |
| EdgeR v3.40.2 | https://doi.org/10.1093/nar/gks042 | |
| UpSetR v1.4.0 R | https://doi.org/10.1093/bioinformatics/btx364 | |
| Factoextra v1.0.7 | https://doi.org/10.32614/CRAN.package.factoextra | |
| EnhancedVolcano v1.16.0 | https://doi.org/10.18129/B9.bioc.EnhancedVolcano | |
| Trinotate | https://doi.org/10.1016/j.celrep.2016.12.063 | |
| VennDiagram 1.7.3 | https://doi.org/10.32614/CRAN.package.VennDiagram | |
| clusterProfiler 4.7.1 | https://doi.org/10.1016/j.xinn.2021.100141 | |
| Ggplot 2 3.4.1 | https://doi.org/10.32614/CRAN.package.ggplot2 | |
| Tidyverse 2.0.0 | https://doi.org/10.21105/joss.01686 | |
| **Other** | | |
| TruSeq stranded sequencing | Illumina | |
| Confocal microscopes | Zeiss | LSM780 or LSM980 |
| Bright-field microscope | Leica | CTR 5000 |
| Cryostar NX70 | Epredia | 957070 |
| Bioanalyzer 2100 | Agilent | G2939A |
| SuperFrost plus glass slides | Epredia | J7800AMNZ |
| Microknifes | Fine Science Tools | 72-2201 |

## Methods and protocols

### *Platynereis dumerilii's culture, amputation procedure and biological material fixation*

*Platynereis* juvenile worms were obtained from a husbandry established at the Institut Jacques Monod (for detailed breeding conditions see (Dorresteijn et al, 1993; Vervoort and Gazave, 2022)). Standard worms used in experiments were 3-4-month-old with 30–40 segments and were amputated according to the procedure described previously (Planques et al, 2019; Vervoort and Gazave, 2022). For the majority of experiments performed (i.e. in situ hybridizations, antibody staining, EdU and TUNEL assays as well as hybridization chain reactions (HCR)—see below for each detailed procedure), regenerative parts at the desired stage and condition were collected and fixed in 4% paraformaldehyde (PFA) diluted in PBS Tween20 0.1% (PBT) for 2 h at room temperature (RT). Following fixation, whole mount samples were washed in PBT, gradually transferred in 100% methanol (MeOH) then stored at −20 °C (Vervoort and Gazave, 2022). For phalloidin staining (see below), after fixation without MeOH dehydration, regenerative

parts were stored in PBT at 4 °C for up to 4 days prior to labelling (Planques et al, 2019).

### Histologic samples fixation and sectioning

Histological sections were performed as described in (Bideau et al, 2024). Briefly, samples were fixed in 4% PFA diluted in PBS 1× for 1h30 at RT, washed in PBS 1×, cryoprotected in a solution of PBS/sucrose 30% for 4–5 days at 4 °C and then transferred into OCT embedding medium (Tissue Freezing Medium, Leica). Next, samples were put into molds and positioned according to the desired type of section (longitudinal), then frozen with dry ice and stored at −80 °C. Samples were cut using a microtome (Leica CM3050S) and sections of 12–14 µM were collected on SuperFrost glass slides prior to storage at −80 °C.

### Whole-mount in situ hybridization (WMISH), antibody staining and phalloidin labelling

Colorimetric NBT/BCIP WMISH and immunolabelling were performed as previously described (Demilly et al, 2013; Vervoort and Gazave, 2022). For all experiments, following rehydration, samples were treated with 40 µg/ml proteinase K in PBT for 10 min, 2 mg/ml glycine PBT for 1 min, 4% PFA PBT for 20 min and washed in PBT prior to hybridization or labelling. A probes list is available in Table EV3. Neurites' labelling was done as previously described (Demilly et al, 2013), using the mouse anti-acetylated tubulin (Sigma 1:500) antibodies and fluorescent secondary anti-mouse IgG Alexa Fluor 488 or 555 conjugate (Cell Signalling, 1:500). For phalloidin labelling, samples were incubated in phalloidin-Alexa 555 (Molecular Probes, 1:100) overnight at 4 °C. Next, samples were nuclei counterstained with Hoechst 0.1% overnight at 4 °C and mounted in glycerol/DABCO (2.5 mg/ml DABCO in glycerol) for confocal imaging (see below). We performed WMISH, neurites and phalloidin labelling on at least five samples for each condition.

### EdU cell proliferation and TUNEL cell death assays

Proliferating cells were labelled by incubating worms with 5 µM of the thymidine analog 5-ethynyl-2′-deoxyuridine (EdU) for 1 h in natural fresh sea water prior to fixation. Various incubation conditions (duration and biological stage) and pulse and chase experiments were performed as described in the Results section and related figures. Fixed samples were subsequently fluorescently labelled with the Click-it EdU Imaging Kit (488 or 555 nm, ThermoFisher) as previously described in (Vervoort and Gazave, 2022). TUNEL labelling was performed using the Click-iT TUNEL kit (647 nm, ThermoFisher), as previously described in (Demilly et al, 2013; Vullien et al, 2025). Briefly, after sample rehydration, cuticle digestion and post fixation, the terminal deoxynucleotide transferase reaction was performed following the kit protocol. In both cases, samples were nuclei counterstained with Hoechst 0.1% overnight at 4 °C and mounted in glycerol/DABCO (2.5 mg/ml DABCO in glycerol) for confocal imaging (see below).

### Hybridization chain reactions (HCR)

HCR coupled with immunolabelling was implemented in *Platynereis* following the HCR 3.0 protocol developed in (Choi et al, 2018). More specifically, the primary antibody incubation of the immunolabelling was performed simultaneously with the amplification step of HCR. Up to 25 couples of probes were designed using

an in-house ProbeMaker based on the HCR 3.0 Probe Maker v1.0 (Kuehn et al, 2022). Each probe sequence was then manually blasted against *Platynereis*'s transcriptome and probes matching several genes were removed. All validated probes were combined in an oligo pool ordered at IDT (see probes list in Dataset EV5). Given the variety of adapters (Molecular Instruments) that can be used with different fluorophores, we made the following selection: *Cdx, Ngn* and *Syt* with the adapter B1 fused with Alexa 647, and *Elav* with the adapter B3 fused with Alexa 594. We performed HCR on a least five samples for each condition.

### Images acquisition, treatments and analyses

Bright-field images of colorimetric WMISH samples were acquired with a Leica CTR 5000 microscope. Fluorescent confocal images of samples/sections were acquired with either a Zeiss LSM780 or LSM980 confocal microscopes. Image processing (contrast and brightness, z-projection, auto-blend layers, transversal and sagittal views) was performed using FIJI and Adobe Photoshop. Figures were assembled with Adobe Illustrator. EdU and TUNEL cell counts were performed using IMARIS 9.5.0 (Oxford Instruments) following the automatic cell counting procedure defined in (Bideau et al, 2024; Vullien et al, 2025). Briefly, for each sample, all nuclei positions (Hoechst + cells) were identified as spots with a standardized nucleus diameter of 5 µm. A region of interest (ROI) corresponding to the regenerative part was then manually delineated, using the Hoechst signal and the general morphology of the structure. Then, the spots inside the ROI were sorted along the fluorescent signals of the EdU or TUNEL labelling. This procedure allowed us to determine the absolute number of nuclei inside the ROI and, among them, the number of positive nuclei for each signal; hence, we could extract the proportions of EdU+ and TUNEL+ for each sample.

Similarly, for measuring cell density of a surface area, we identified all nuclei positions, manually delineated a ROI and determine the surface of the structure, then counted the spots inside the whole ROI. The density was defined as the number of nuclei per µm² for the whole structure.

### Treatments with γ-secretase inhibitors, scoring and statistical analyses

Chemical inhibitions of the Notch signalling pathway were performed using three different inhibitors, DAPT, LY-411575 and RO-4929097, widely used to specifically block the gamma-secretase complex responsible, while not exclusively, for the cleavage of Notch, thus preventing transcription of the target genes (Golde et al, 2013). Gamma-secretase is a multi-subunit protease complex consisting of one proteolytically active subunit, presenilin (PS), and three non-proteolytic subunits nicastrin, APH-1 (anterior pharynx defective-1) and presenilin enhancer 2 (PEN-2) (Appendix Fig. S1). Gamma-secretase plays a critical role in the cleavage of several membrane proteins or substrates, including the Notch receptors, but also amyloid precursor protein (APP), DCC and Ephrin receptors among others (see (Guner and Lichtenthaler, 2020) for a recent list of substrates). These inhibitors have previously been successfully used to disrupt Notch signaling in *Platynereis* embryos and larvae (Gazave et al, 2017), but also in a diversity of organisms during both development and regeneration (Dray et al, 2021; Gahan et al, 2017; Grotek et al, 2013; Hamada et al, 2015; Mashanov et al, 2020; Munch et al, 2013; Munder et al, 2013). We first determined the efficient concentrations for each inhibitor by performing treatments with different concentrations (1, 5 and 10 µM for LY-411575, 5 and 10 µM for RO-4929097 and 40 µM for DAPT, based on (Gazave et al, 2017), from a 10 mM stock solutions in DMSO and in comparison to DMSO controls). Upon amputation, we immediately incubated the worms in 2 ml of each inhibitor solution on 12-well plate and assessed the effects by scoring the regenerative stages reached by each worm every day for 5 days of treatment (when posterior regeneration is over) (Planques et al, 2019; Vervoort and Gazave, 2022). Control worms were incubated in natural fresh sea water with identical concentrations of DMSO. All solutions were refreshed every 24 h to maintain their activities for the whole duration of the experiments (Vervoort and Gazave, 2022). Thus, 5 µM for both LY-411575 and RO-4929097 and 40 µM for DAPT, were found to be the most effective concentrations, resulting in a consistent and reproducible effect: blocking regeneration around stage 2 (Appendix Fig. S2A–C, respectively). As all inhibitors led to similar effects on regeneration, we decided to pursue our experiments using only LY-411575 at 5 µM.

All statistical tests and subsequent graphical representations were performed using GraphPad Prism 9. Mann–Whitney $U$ tests were used to compare samples between different experiments/conditions. $P$ values are indicated, for each test, directly on the figures.

### Determination of gene expression levels during posterior regeneration

In an updated version of our *Platynereis* reference transcriptome (Paré et al, 2023) (Table EV1), 26 genes corresponding to both the Notch signaling pathway core machinery and their putative target genes from the Hairy enhancer of split multigenic family (Gazave et al, 2014; Gazave et al, 2017) were identified. Their respective expression levels were determined during the course of regeneration (i.e. stages 0, 1, 2, 3, 5 days post amputation as well as non-amputated control – 2 to 3 replicates) using our previously produced RNA-seq datasets of posterior regeneration (Paré et al, 2023) (Dataset EV1). Expression level dynamics were visualized using heatmap.2 from the gplots R package.

### Transcriptomic analysis of Notch pathway inhibition effects on posterior regeneration

*Sample production and collection*: 1 dpa and 2 dpa samples treated with 5 µM of LY-411575 and 0.05% DMSO controls were produced. For each stage and condition, three biological replicates per stage and condition were produced independently, each one containing 200 regenerating parts recovered with as little non-amputated tissue as possible (typically half a segment).

*RNA extraction, library construction and sequencing*: For all samples ($n = 12$), total RNA was extracted and its quality was assessed as described previously (Paré et al, 2023). Libraries and Illumina TruSeq Stranded sequencing (75 bp in single-end) were performed at the Ecole Normale Supérieure GenomiqueENS core facility (Paris, France), as detailed in (Paré et al, 2023). All raw reads from individual sequencing libraries are deposited in the European Nucleotide Archive (ENA) (Table EV1).

*Read processing and mapping*: Reads were quality checked using FastQC v0.11.8 and trimmed for low quality reads and adapter using fastp v0.23.2. Kallisto v0.48.0 within the Trinity v2.13.2 toolkit (Haas et al, 2013) was then used to perform pseudo-mapping and quantification of the reads on the updated reference transcriptome and to generate the raw count matrix. The raw count

matrix was processed using EdgeR v3.40.2 (McCarthy et al, 2012) to obtain a TMM count matrix. The intersection of expressed genes between each condition was plotted using the UpSetR v1.4.0 R package (Conway et al, 2017). PCA plots were performed based on the processed raw count matrix to a count per million matrix. Only genes with a TMM value superior or equal to 1 TMM were retained. The count per million matrix was batch-corrected using the removeBatchEffect function from EdgeR. The PCA plot was performed using the factoextra R package v1.0.7.

*Differentially expressed genes identification*: For each treated *versus* control condition (at 1 and 2 dpa), differential expression analyses were conducted using the EdgeR R package based on the raw count matrix. Only genes with a TMM value superior or equal to 1 TMM in at least one condition were considered. A design matrix was built to perform batch correction. The *P* values were calculated using the likelihood ratio test and adjusted for multiple tests using the Benjamini–Hochberg FDR correction. Genes with an adjusted *P* value less than 0.05 and an absolute log fold change value of at least 1 were considered as differentially expressed genes (DEGs). Volcano plots were constructed using the EnhancedVolcano v1.16.0 R package (Blighe et al, 2021). DEGs were annotated with Trinotate (Bryant et al, 2017) and from the top homology blast on the mouse proteome, as previously described (Paré et al, 2023). We used the VennDiagram package in R (Conway et al, 2017) to quantify and visualize shared DEGs between comparisons.

*Gene ontology and enrichment analysis for differentially expressed genes*: We performed Gene Ontology (GO)-term enrichment analyses on DEG lists using clusterProfiler (Wu et al, 2021). For the DEGs, a full list of enriched GO terms is provided in Datasets EV2, 3 and the top 20 per comparison for the category "Biological processes" are presented on dotplots.

## Data availability

The sequencing data generated in this project have been deposited at the European Nucleotide Archive (ENA) repository under the project accession number PRJEB63219. The scripts used in this project are available on Zenodo https://doi.org/10.5281/zenodo.18341641 or at https://github.com/StemDevEvo/Notch-2025. The original confocal and bright field images have been deposited at the BioImage Archive with accession number S-BIAD2428.

The source data of this paper are collected in the following database record: biostudies:S-SCDT-10_1038-S44319-026-00731-6.

## Peer review information

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

## Acknowledgements

We are grateful to all past and present members of the Gazave lab for their support and suggestions on this study. We thank the Institute Jacques Monod facility staff for their help with the *Platynereis* culture. We acknowledge the ImagoSeine core facility of Institut Jacques Monod, member of France-BioImaging (ANR-10-INBS-04) and IBiSA, with the support of Labex "Who Am I", Inserm Plan Cancer, Region Ile-de-France and Fondation Bettencourt Schueller. Work in our team is supported by funding from: Labex "Who Am I" laboratory of excellence (No. ANR- 11-LABX-0071) funded by the French Government through its "Investments for the Future" program operated by the Agence Nationale de la Recherche under grant No. ANR-11-IDEX-0005-01, Agence Nationale de la Recherche «STEM» (ANR-19-CE27-0027-01)), Centre National de la Recherche Scientifique (CNRS), INSB (Grant Diversity of Biological Mechanisms), Université Paris Cité, Association pour la Recherche sur le Cancer (grant PJA 20191209482), and comité départemental de Paris de la Ligue Nationale Contre le Cancer (grant RS20/75-20). LB has obtained a CDSN PhD fellowship from ENS Lyon and his fourth year of PhD was supported by the Labex "Who am I". The GenomiqueENS core facility is supported by the France Génomique national infrastructure, funded as part of the "Investissements d'Avenir" program managed by the Agence Nationale de la Recherche (contract ANR-10-INBS-0009).

## Author contributions

**Loïc Bideau**: Conceptualization; Formal analysis; Validation; Investigation; Visualization; Methodology; Writing—original draft; Project administration; Writing—review and editing. **Loeiza Baduel**: Validation; Investigation; Methodology. **Gabriel Krasovec**: Investigation; Writing—original draft; Writing—review and editing. **Caroline Dalle**: Investigation; Methodology. **Ombeline Lamer**: Investigation. **Mélusine Nicolas**: Investigation. **Alexandre Couëtoux**: Investigation. **Corinne Blugeon**: Resources; Data curation; Methodology. **Louis Paré**: Data curation; Software; Formal analysis; Investigation; Visualization; Methodology; Writing—review and editing. **Michel Vervoort**: Conceptualization; Resources; Supervision; Funding acquisition; Project administration. **Pierre Kerner**: Conceptualization; Investigation; Writing—original draft; Writing—review and editing. **Eve Gazave**: Conceptualization; Resources; Data curation; Formal analysis; Supervision; Funding acquisition; Validation; Investigation; Visualization; Methodology; Writing—original draft; Project administration; Writing—review and editing.

Source data underlying figure panels in this paper may have individual authorship assigned. Where available, figure panel/source data authorship is listed in the following database record: biostudies:S-SCDT-10_1038-S44319-026-00731-6.

## Disclosure and competing interests statement

The authors declare no competing interests.

EMBO reports

Loïc Bideau et al

# Expanded View Figures

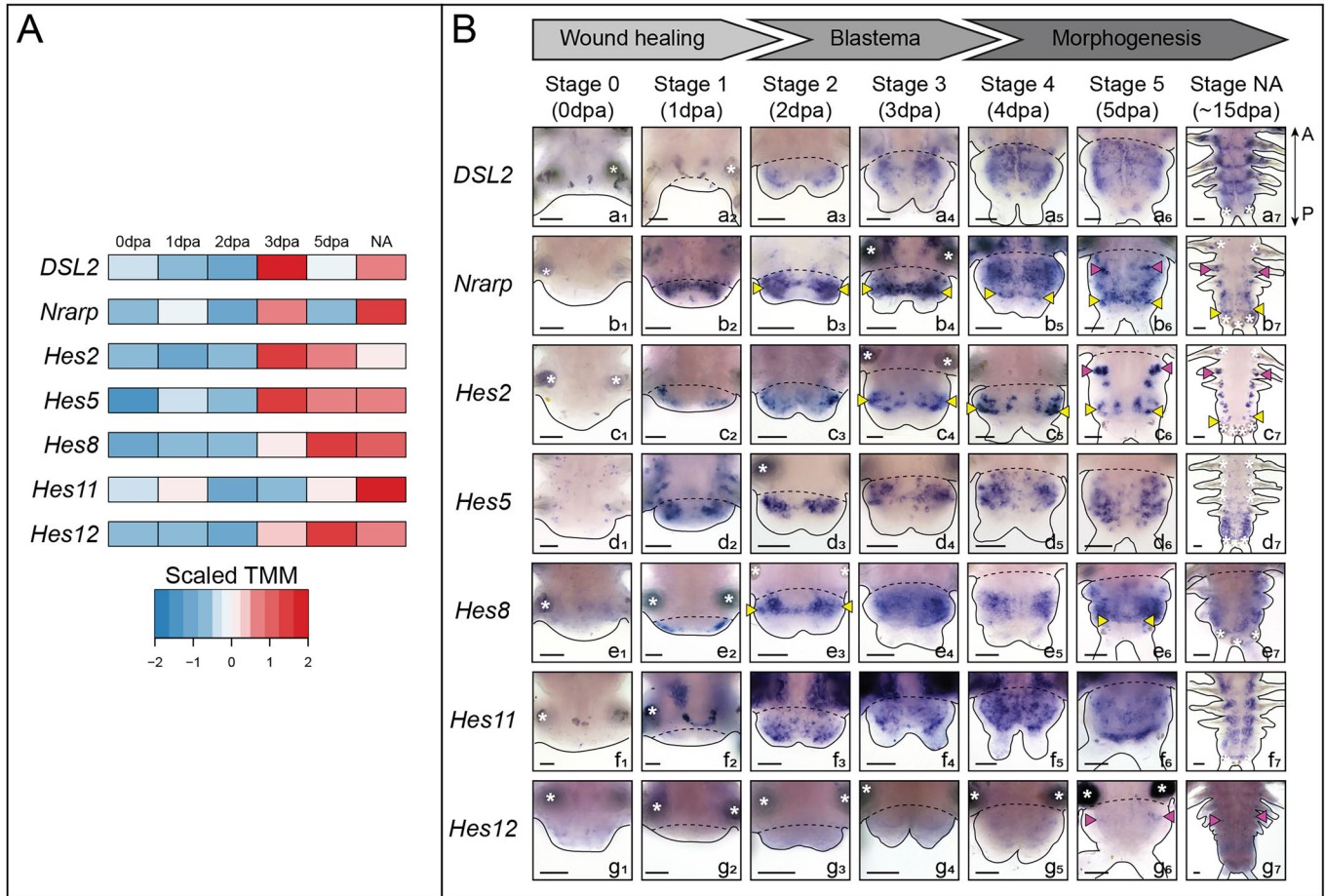

**Figure EV1. Dynamic expression of core members of the Notch pathway and its putative target genes in non-neurogenic territories during posterior regeneration.**

(A) Heatmap representation of expression levels of several Notch components and *Hes* genes during posterior regeneration (Paré et al, 2023). (B) Whole-mount in situ hybridizations (ventral views) of Notch components and *Hes* genes expressed in non-neurogenic structures during regeneration. Data information: yellow arrowheads = growth zone involved in posterior elongation of the animals (Gazave et al, 2013); pink arrowheads = chaetal sacs producing the parapodial bristles; white asterisks = non-specific staining from glands. dpa = day(s) post-amputation, NA = non-amputated. Solid black lines delineate the outlines of the samples, black dashed lines correspond to the amputation planes. Scale bars = 50 μm. Anteroposterior (A/P) axis is represented. All images come from representative samples of at least two biological replicates. Source data are available online for this figure.

2364    EMBO reports    Volume 27 | May 2026 | 2345 – 2368

© The Author(s)

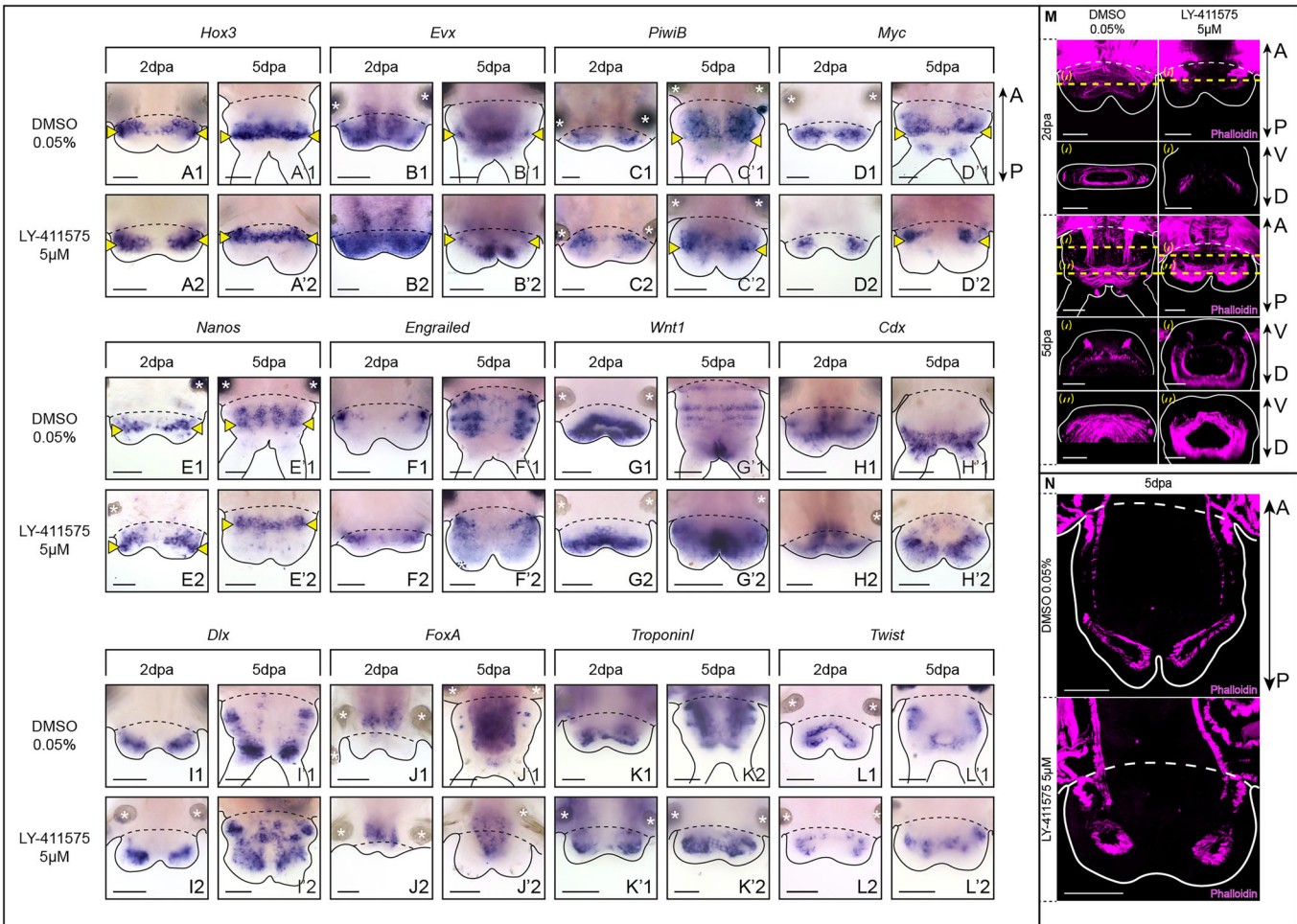

**Figure EV2.    Effects of Notch signaling pathway inhibition on several tissues during posterior regeneration in *Platynereis*.**

(A–L) Whole-mount in situ hybridizations for markers of the growth zone (A, B), stem cells (C–E), segmentation (F, G), pygidium (H), pygidial cirri and appendages (I), gut (J) and muscles (K, L) for LY-411575 treated worms and controls at 2 and 5 dpa. Ventral views. Solid black lines delineate the outlines of the samples, black dashed lines correspond to the amputation planes. (M) Phalloidin labelling on whole-mount regenerated parts of LY-411575-treated worms and DMSO controls at 2 and 5 dpa. Ventral views are on top and corresponding virtual transverse sections (along the yellow dotted lines) are at the bottom. (N) Phalloidin labelling on longitudinal cross-sections of LY-411575-treated worms and controls at 5 dpa. (M, N) Solid white lines delineate the outlines of the samples, white dashed lines correspond to the amputation planes. Yellow arrowheads = growth zone involved in posterior elongation of the animals (Gazave et al, 2013); white asterisks = non-specific staining from parapodial glands. dpa = day(s) post-amputation. Scale bars = 50 µm. Anteroposterior (A/P) and dorsoventral (D/V) axes are represented. All images come from representative samples of at least two biological replicates. Source data are available online for this figure.

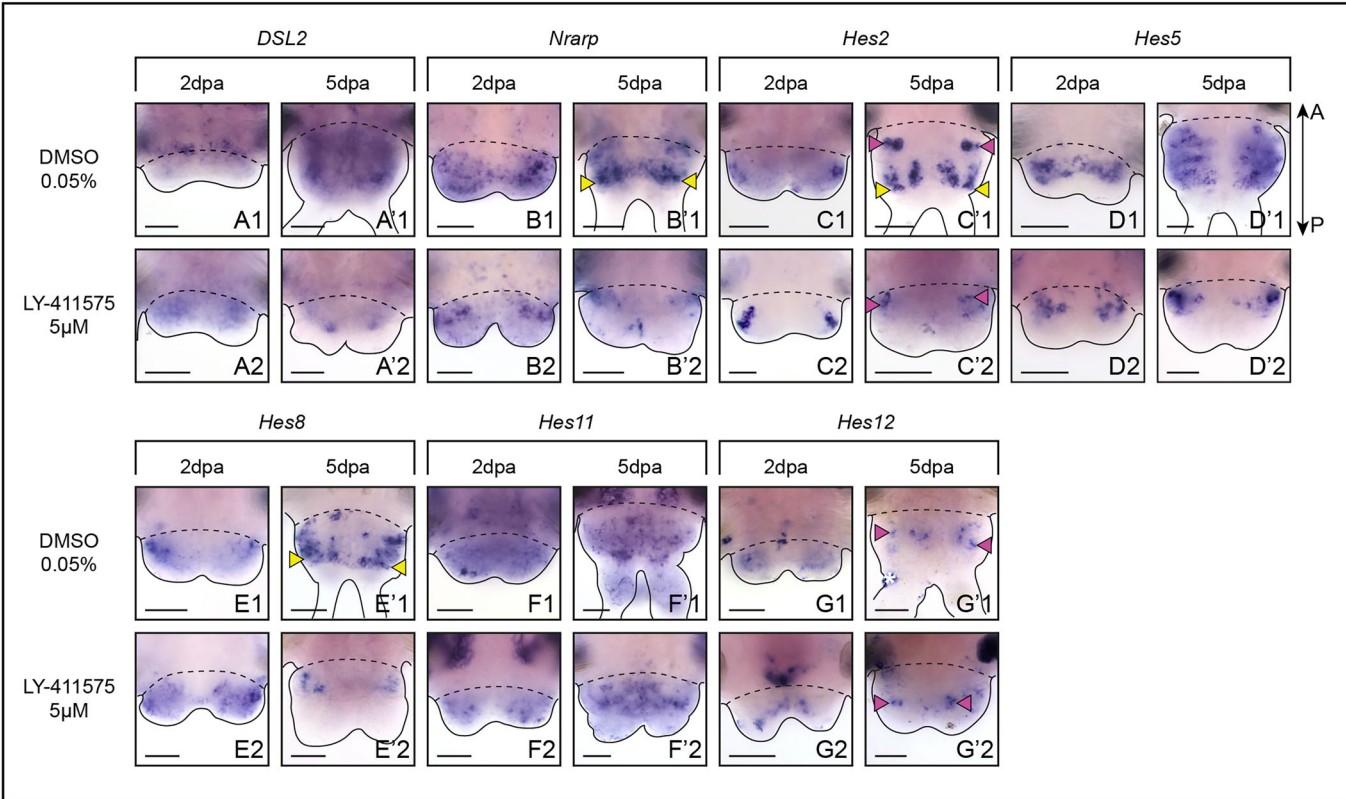

**Figure EV3.  Impact of Notch pathway inhibition on non-neural territories during posterior regeneration.**

(A–G) Whole-mount in situ hybridizations of Notch components and *Hes* genes expressed in non-neurogenic structures for LY-411575 treated worms and DMSO controls at 2 and 5 dpa. Ventral views. Solid black lines delineate the outlines of the samples, black dashed lines correspond to the amputation planes. Yellow arrowheads = growth zone involved in posterior elongation of the animals (Gazave et al, 2013); pink arrowheads = chaetal sacs producing the parapodial bristles; white asterisks = non-specific staining from glands. dpa = day(s) post-amputation. Scale bars = 50 μm. Anteroposterior (A/P) axis is represented. All images come from representative samples of at least two biological replicates. Source data are available online for this figure.

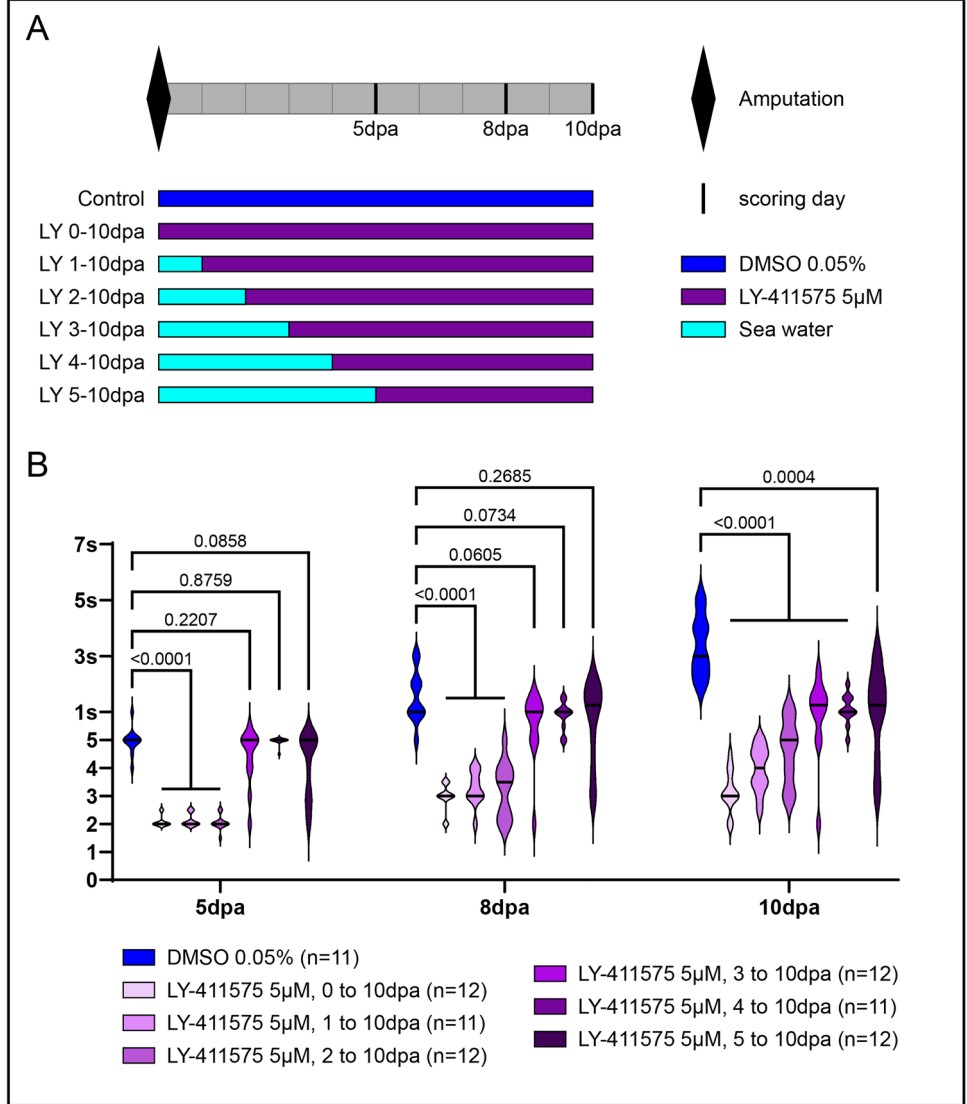

**Figure EV4. Morphological effects of different durations of Notch pathway inhibition along posterior regeneration and posterior elongation.**

(**A**) Schematic representation of the experiments: six durations of LY-411575 treatment were performed as well as a DMSO control. (**B**) Violin plots representing the stages reached by each worm at 5, 8 and 10 dpa for each treatment. "*n*" represents the number of worms used per condition (*n* ranging from 11 to 12). Data in (**B**) are representative of two independent experiments and unpaired Mann–Whitney *U* tests were used for statistical analyses. *P* values are indicated in the figure and in the Table EV4. dpa = day(s) post-amputation. Source data are available online for this figure.

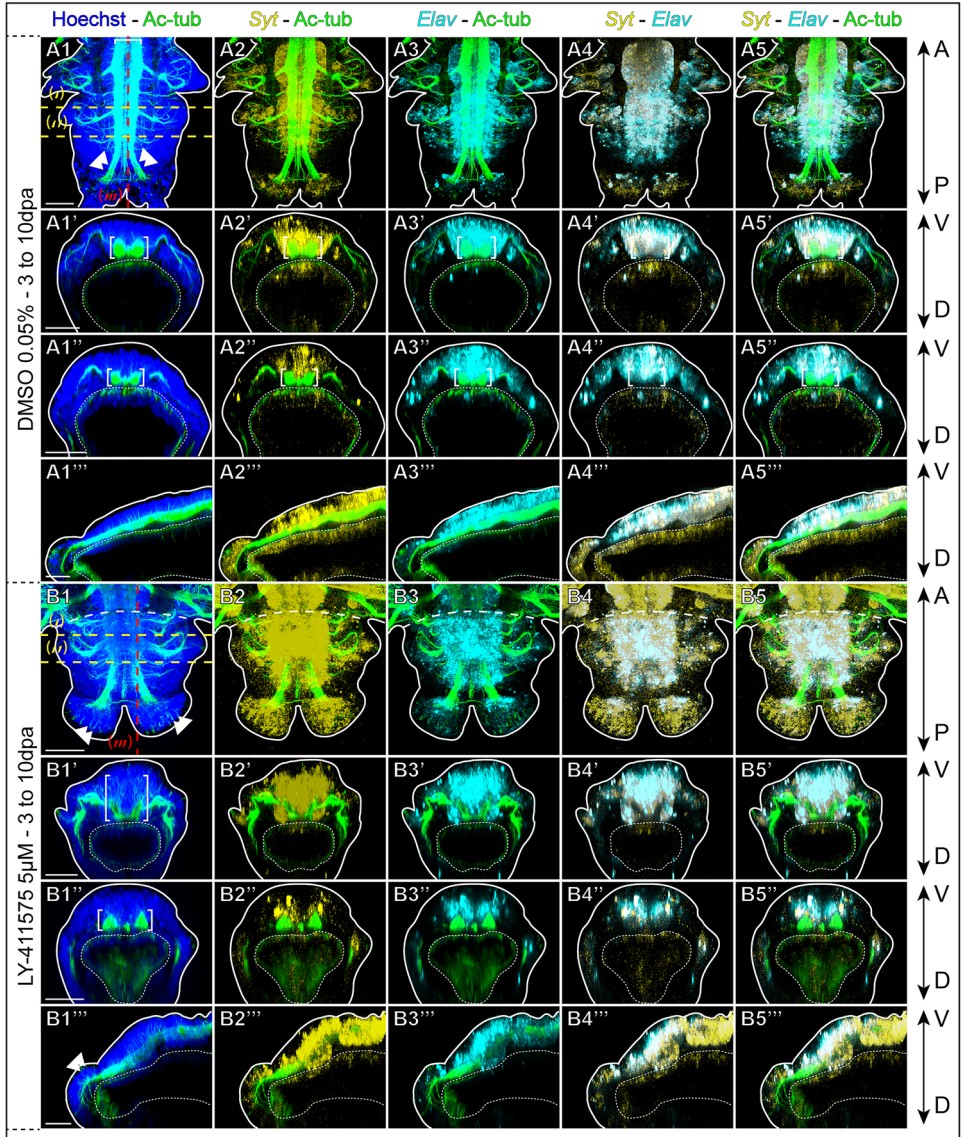

**Figure EV5. The neuronal marker *Syt* confirms the phenotype obtained upon Notch pathway inhibition during post-regeneration posterior elongation.**

Hybridization chain reactions (HCR) for *Elav* (cyan) and *Syt* (yellow) coupled with immunolabelling for acetylated tubulin (green) and nuclei staining with Hoechst (blue) for controls at 10 dpa (top, **A**) and LY-411575 treated regenerated parts from 3 dpa to 10 dpa (bottom, **B**). Ventral views are on top for each condition and corresponding virtual transverse sections (along (') and ('') in yellow) and sagittal section (along (''') in red) are at the bottom. Solid white lines delineate the outlines of the sample, and white dotted lines delineate the gut. White brackets = ventral nerve cord; white double arrowheads = cirri nerves; blue arrowheads = PNS. dpa = day(s) post-amputation. Scale bars = 50 µm. Anteroposterior (A/P) and dorsoventral (D/V) axes are represented. All images come from representative samples of two biological replicates. Source data are available online for this figure.

