## [Peer Review File · EMBO Reports]

Multifaceted conserved functions of Notch during adult neurogenesis in the annelid *Platynereis*

Loïc Bideau, Loeiza Baduel, Gabriel Krasovec, Caroline Dalle, Ombeline Lamer, Mélusine Nicolas, Alexandre Couetoux, Corinne Blugeon, Louis Paré, Michel Vervoort, Pierre Kerner, and Eve Gazave

Corresponding author(s): Eve Gazave (eve.gazave@ijm.fr)

Review Timeline:

Submission Date:	15th Apr 25
Editorial Decision:	30th May 25
Revision Received:	25th Nov 25
Editorial Decision:	20th Jan 26
Revision Received:	27th Jan 26
Accepted:	12th Feb 26

Editor: Yehu Moran

Transaction Report:

Dear Dr. Gazave

Thank you for the submission of your manuscript to EMBO Reports. We have now received the full set of referee reports as well as referee cross-comments that are all pasted below.

As you will see, the referees acknowledge that the findings are potentially interesting. However, they do raise some significant concerns that require your attention.

I would thus like to invite you to revise your manuscript with the understanding that the referee concerns must be fully addressed and their suggestions taken on board. Please address all referee concerns in a complete point-by-point response. I would like to ask you to put special emphasis on the concern raised by multiple referees that the gamma-secretase inhibitor you used in your experiments might affect additional pathways beyond Notch.

Acceptance of the manuscript will depend on a positive outcome of a second round of review. It is EMBO Reports policy to allow a single round of major revision only and acceptance or rejection of the manuscript will therefore depend on the completeness of your responses included in the next, final version of the manuscript.

We realize that it is difficult to revise to a specific deadline. In the interest of protecting the conceptual advance provided by the work, we recommend a revision within 3 months (30th Aug 2025). Please discuss the revision progress ahead of this time with the editor if you require more time to complete the revisions.

- 1) A data availability section providing access to data deposited in public databases is missing. If you have not deposited any data, please add a sentence to the data availability section that explains that.
- 2) Your manuscript contains statistics and error bars based on $n=2$. Please use scatter blots in these cases. No statistics should be calculated if $n=2$.

5) a complete author checklist, which you can download from our author guidelines

<<https://www.embopress.org/page/journal/14693178/authorguide>>. Please insert information in the checklist that is also reflected in the manuscript. The completed author checklist will also be part of the RPF.

6) Please note that all corresponding authors are required to supply an ORCID ID for their name upon submission of a revised manuscript (<<https://orcid.org/>>). Please find instructions on how to link your ORCID ID to your account in our manuscript tracking system in our Author guidelines <<https://www.embopress.org/page/journal/14693178/authorguide#authorshipguidelines>>

7) Before submitting your revision, primary datasets produced in this study need to be deposited in an appropriate public database (see <https://www.embopress.org/page/journal/14693178/authorguide#datadeposition>). Please remember to provide a reviewer password if the datasets are not yet public. The accession numbers and database should be listed in a formal "Data Availability" section placed after Materials & Method (see also <https://www.embopress.org/page/journal/14693178/authorguide#datadeposition>). Please note that the Data Availability Section is restricted to new primary data that are part of this study. * Note - All links should resolve to a page where the data can be accessed. *
If your study has not produced novel datasets, please mention this fact in the Data Availability Section.

12) All Materials and Methods need to be described in the main text using our 'Structured Methods' format, which is required for all research articles. According to this format, the Methods section includes a Reagents and Tools Table (listing key reagents, experimental models, software and relevant equipment and including their sources and relevant identifiers) followed by a Methods and Protocols section describing the methods using a step-by-step protocol format. The aim is to facilitate adoption of the methodologies across labs. More information on how to adhere to this format as well as a downloadable template (.docx) for the Reagents and Tools Table can be found in our author guidelines: <https://www.embopress.org/page/journal/14693178/authorguide#structuredmethods>.

An example of a Method paper with Structured Methods can be found here: <https://www.embopress.org/doi/full/10.1038/s44320-024-00037-6#sec-4>

I look forward to seeing a revised form of your manuscript when it is ready.

Yours sincerely,

Yehu Moran
Academic Editor
EMBO Reports

Referee #1:

In their manuscript Bideau et al attempt to understand the role of Notch signalling during regeneration and neurogenesis in *Platyneris*. Although Notch signalling has been well studied in many "established" models its role across animal diversity are not well understood nor are the evolutionary origins of many of those roles. The authors are therefore attempting to address an important and timely question. Overall the quality of data presented by the authors is very high with well annotated images. Several of the detailed explanations on morphology are difficult to follow for a non-expert on the animal but the authors do a good job of annotating images and leading the reader through their analysis. I do however have a number of criticisms of the manuscript as outlined below.

1. The authors rely entirely on the use of γ -Secretase inhibitors as a proxy for Notch inhibition. Although Notch is a well studied target for gamma-secretase it is by no means the only protein cleaved by γ -Secretase. In particular in the context of later stages of neurogenesis there have been several relevant substrates identified including DCC which has known roles in neurite outgrowth. This reliance on indirect inhibition of the pathway is a major concern for all the findings in the manuscript. In the absence of direct functional analyses, it is therefore impossible for the authors to clearly assign any phenotype to Notch signalling. Statement like "we definitively demonstrated that the Notch pathway regulates neurogenesis during both pygidial regeneration and CNS patterning during posterior elongation" are therefore incorrect as this has not been definitely shown by the set of experiments in the paper.
2. The authors state, based on the data in Figure 4, that "these findings demonstrate that Notch signaling acts as a key regulator of pygidial neurogenesis by controlling neural progenitor specification". This is not well supported by the data. Even if we assume that the inhibitor is specific to Notch signalling, it cannot be deduced from the data what is the proximal cause of the observed phenotypes. Notch signalling could be playing a role in neural progenitors but equally any of the other tissue-level defects that have been described earlier could be secondarily leading to the effects on the nervous system.
3. The next section in title "Notch regulates pygidial neurogenesis through the ligand Delta and several Hes target genes". In this case there is again no evidence on the roles of Delta and Hes genes. The authors show the inhibitors have an effect on these genes but whether the observed effects are "through" the activities of these genes is not explored at all in the manuscript. This would require a lot of functional work on Delta and Hes genes.
4. In their experiments to study the later effects of inhibitors the authors do not discuss the rather larger tissue level differences between control and treated animals. A more thorough explanation of this would be wellcome

Minor comments:

1. In figure 2, the Authors state that there is no difference in the "distribution of Edu+ cells" but to me the images shown show some drastic differences in the localization of the Edu+ cells, most strikingly in the bottom of Panel 2 where the side with Edu+ cells seems to have completely switched. The same is true in Supp. Fig2D. The authors should attempt to quantify the position of cells before stating that there is no difference in distribution.
2. Based on Supp Fig 3K-M the authors state that "proper gut and muscle regeneration" but the images suggest that it is rather abnormal and perhaps "proper" is not the correct term.
3. In fig. 3A the multiple colours are confusing. It would be better to make just red genes that are "DEGs" and the rest all grey. It is not clear in the figure what the gene names point to...i.e. they are not assigned to a particular point in the plot. I would either put them assigned to a point or just mention the names in the text.
4. The numbers in Fig.3A' do not add up. Based on the venn diagram there are 698 genes differentially expressed (313+385) at 1dpa but this is less than shown above. The same for 2dpa genes.
5. In figure 6, the DMSO and treated animals are not labelled like in other figures.

Referee #2:

Summary & Significance

This valuable manuscript of Bideau et al. studies Notch pathway-dependent pygidium regeneration in the juvenile stage of the marine annelid *Platynereis dumerilii*. The work focuses on amputation-induced regeneration, neurogenesis and morphological and transcriptional comparisons in control animals and animals with inhibited Notch signalling. The authors concluded that Notch signalling has multiple conserved functions in juvenile *Platynereis* neurogenesis. The authors present strong evidence to support their conclusion that Notch signalling regulates neurogenesis during posterior regeneration although some concerns remain about the specificity of the inhibitor treatments. The study will be of interest to developmental biologists working on Notch signalling and regeneration.

Strengths

The authors use a clear tail regeneration paradigm and combine it with extensive gene expression analyses and inhibitor treatments. They analyse several Notch signalling-related genes both by high-quality in situ hybridisations and RNA-seq approaches at different stages of regeneration. The authors also quantify cell proliferation, cell death and nervous system morphology. The data are in general of high quality.

Weaknesses

The authors mostly used LY-411575 inhibition to block the Notch pathway. This compound is a very potent γ -secretase inhibitor that inhibits Notch cleavage at low nanomolar concentrations. The authors used 5 micromolar concentration in their experiments. The inhibitor is known to also affect other pathways that depend on γ -secretase activity and at the high concentrations used other side effects may also occur. The authors should carefully consider and discuss these potential side effect and also check in their RNAseq data if other pathways are effected. The authors should also include a more detailed explanation of how the inhibitor acts and what are its potential targets.

Minor Comments

Writing style: The manuscript in some sections lacks clarity and there are inconsistencies in the terminology. A detailed list of suggestions is given below.

The authors could consider including in situs for Jagged and Presenilin and discuss their expression dynamics after amputation (based on their RNAseq data these genes change their expression and they are part of the pathway)

The authors should include a picture of the full juvenile worm highlighting its growth zone and also a summary of the experimental manipulations (position of cuts) and design and the Notch pathway with its core components.

at least once per figure indicate the orientation of the specimens with arrows; at least once per figure include a scale bar;

mark the amputation site whenever possible to have a better reference of the relative anatomy

the methods section should include a data and code availability statement with DOI of the repository

source data for all plots should be provided

please include a more detailed protocol for TUNEL staining and provide all antibody specifications

please provide more information about the conventional WMISH probes, including plasmid map and availability or PCR primers, probe length

Page5 Line14: briefly explain how the previous RNA dataset was generated (e.g. stages, manipulation)

please indicate the number of individuals that were compared per condition in the amputation and in situ experiments

SupFig7: please mention if the seawater (cyan) also contains 0.05% DMSO as the control condition at the same time points

Language:

P4L4: clarify terminology of cirri, antennae, etc.
P4L7: mention that HES genes are members of the bHLH family
P4L22: maintain the listing format (Third, ...) for easier read
P5L2: mention pygidium earlier when introducing regeneration of body parts.
the term "growth zone (GZ)" and "segment addition zone" (SAZ) are used in the literature, please mention this or explain the difference
Vesicular acetylcholine transporter -should be VACHT
"Notch inhibition" should rather be written as Notch-pathway inhibition because it is not Notch that is directly inhibited
P5L22: "Strikingly" - avoid words like "Interestingly" or "Strikingly"
P6L1: which six core members & which four are expressed? mention the core members in the introduction
Fig1: NA here means: "healthy" non-amputated control (Ref35), but in B NA means 15 days after amputation
P7L21: Shortly state that worms were treated immediately after amputation.
P7L18: describe the hypertrophied tissue in more detail e.g. more extended in the ventral-dorsal axis (round shape) compared to a flat oval shape in the non treated animals.
P9L1: only use "significant" if statistical test were done
P8L18: What does "few tissues" exactly mean?, "hypertrophic proliferative pygidium"
P14L23: "Superficially is a highly proliferative Ngn+ expressing layer of neuronal progenitors" - use a more precise term than "superficially" e.g. close to the dorsal ectoderm etc
Fig6: switch the position order from Collier and Elav to the order you depicted in Fig4, Elav and Collier.
Fig7: In A the virtual cross sections (' & ') are too wide compared to the ventral view and thereby not in proper scale. (see Panel B).
Fig7: The longitudinal and cross sections are contradicting each other as the longitudinal section depicts all three markers to be present up to the top layer of the specimen
SupFig2: lnE: n is missing; $\mu\text{m}^2 > \mu\text{m}^2$
SupFig3: align numbering; K2: pygidium seems to be present at 2 dpa although amputated and gone in K'2;
SupFig4: the black and white schematic is not 100% clear on first sight, maybe add a small description/color parts of the animal to clarify the difference.
Fig6: add treatment & stage of the depicted images; use comparable specimen regions for the stainings; add amputation lines in the in situ stainings. Add description of blue arrowheads.

Typos

"prefiguring longitudinal tracks of different neuron types." tracts
"the addition of newly formed segment at their posterior end" -segments
"regulates both the neural progenitor determination and the differentiated neurons balance in the regenerated pygidium" the determination of neural progenitors and the balance of differentiated neurons
The notation "that Platynereis' genome" etc is a bit odd, it reads better as 'the Platynereis genome' or 'the genome of Platynereis'
"are not merely arrested at approximately stage 2, but rather exhibits" rather exhibit
"the two nerves innervating the anal cirri" these nerves rather contain nerves that come from the anal cirri sensory neurons and run into the nerve cord so better to say the two nerves running from the anal cirri or simply the two nerves of the anal cirri
"that early chemical inhibition of Notch pathway" of the Notch pathway
"Notch signaling controls pygidial neurogenesis during regeneration by regulating Hes genes activity" Hes-gene activity or the activity of Hes genes
"Given the dramatic nervous system defects induced by Notch inhibition during posterior regeneration, we decided to further explore its impact on gene expression thanks to a bulk RNA-seq unbiased approach between LY-411575-treated and control worms at 1 and 2 dpa (Sup. Fig. 4A and B, Supp. Table 2)." reword - thanks to does not work here
"and ectopic neurons formation" ectopic neuron formation
"Importantly, one fifth of DEG are related to nervous system," here specify how many of these are up or down regulated
"salt and paper fashion" pepper
"Notch signalling pathway inhibitors treatments" Treatments with Notch signalling pathway inhibitors
"Supp. Fig. 4" Sup. Fig. 4

Referee #3:

The manuscript presents a significant contribution to our understanding of Notch signalling in bilaterian neurogenesis. The study provides valuable insights into the conserved role of Notch in regulating neural progenitors and neuron fate in the pygidium. Furthermore, the findings are relevant for the Spiralia community, enhancing our comprehension of both regeneration and neurogenesis during growth processes. The results are novel, significant, and clearly stated. The data presented support the conclusions effectively, and the comprehensive methodology employed adds robustness to the findings. The integration of diverse experimental techniques is particularly commendable.

I would like to recommend the manuscript with minor revision. Only minor issues are raised that will further enhance clarity and precision.

Minor Suggestions:

Figures:

Consider excluding figure descriptors within the text (e.g., change "Fig.1B - green square bracket" to just "Fig.1B"), as this information is clearly presented in the figure legends.

Ensure that figure panels are consistently labelled in the text (e.g., "Supp. Fig. 1B-e, f").

Figure consistency by i) using uniform panel labels (e.g., in Fig. 6, the terms "contr" and "inhibition" should be added), ii) ensuring all arrowheads are clearly visible/ included, and iii) increasing arrowhead size in the main figures is recommended, as well as iv) consistent alphabetical labels (e.g., A, B, C...) across supplementary figures (e.g., S-fig 5).

Text:

The manuscript is well written and clear, with thorough and well-conducted experiments.

Ensure consistency in terminology (e.g., "anal" vs. "pygidial cirri").

In the sentence: "To further characterize this phenotype, we conducted an extensive molecular analysis using a set of markers to label different structures and tissues involved in Platynereis' posterior regeneration (Fig. 2, Supp. Fig. 3) [28, 29, 36]" - Please clarify the main figure numbers, as they seem currently unclear.

Methods:

Please include in the Method part details on how EdU+ labelled cells and Tunnel+ cells were counted in Fig. 2C.

Similarly, specify how the density of the surface area was measured in S-Fig 2. It would be helpful to clarify whether a single plane was analysed.

We want to thank the referees for their positive and constructive comments, which have significantly improved our work. We are pleased to provide a revised manuscript that addresses all the comments raised. The main changes to our manuscript are summarized as follows:

- We took into consideration the concerns raised by two referees that the γ -secretase inhibitor LY-411575 we used in our experiments might affect additional pathways beyond Notch, one of which is DCC/Netrin pathway that plays important roles in axon guidance. Therefore, we performed additional experiments using a specific inhibitor (PP2) of a major pathway downstream of DCC/Netrin (*i.e.* Src family kinases or SFKs). We then compared the phenotypes obtained with LY-411575 and PP2. We also analysed our RNA-seq data to check for differentially expressed genes that could be linked to DCC/Netrin or other signalling pathways. The results obtained strengthened our initial findings. However, as it is technically impossible to generate gene knockouts in our model system in regeneration, we toned down some of our statements and included a specific paragraph in the method section about the substrates of the γ -secretase inhibitor LY-411575. We also added a 'Limitations of the study' section which develops all these points.
- We made ectoderm *versus* meso-endoderm counting for EdU+ cells in treated *versus* control individuals and highlighted a differential distribution of the EdU+ cells upon LY-411575 treatment.
- We made additional ISH for *Presenilin* and tissue-specific markers at later stages and discussed larger tissue-level defects.

Below, we provide a detailed point-by-point response to each referee's concerns. We hope we have addressed them satisfactorily and will address any additional comments or suggestions that may be raised.

Referee #1:

In their manuscript Bideau et al attempt to understand the role of Notch signalling during regeneration and neurogenesis in *Platyneris*. Although Notch signalling has been well studied in many "established" models its role across animal diversity are not well understood nor are the evolutionary origins of many of those roles. The authors are therefore attempting to address an important and timely question. Overall the quality of data presented by the authors is very high with well annotated images. Several of the detailed explanations on morphology are difficult to follow for a non-expert on the animal but the authors do a good job of annotating images and leading the reader through their analysis. I do however have a number of criticisms of the manuscript as outlined below.

We thank the referee for this positive appreciation of our work. We have addressed all the comments raised, as detailed below.

1. The authors rely entirely on the use of γ -Secretase inhibitors as a proxy for Notch inhibition. Although Notch is a well studied target for gamma-secretase it is by no means the only protein cleaved by γ -Secretase. In particular in the context of later stages of neurogenesis there have been several relevant substrates identified including DCC which has known roles in neurite outgrowth. This reliance on indirect inhibition of the pathway is a major concern for

all the findings in the manuscript. In the absence of direct functional analyses, it is therefore impossible for the authors to clearly assign any phenotype to Notch signalling. Statement like "we definitively demonstrated that the Notch pathway regulates neurogenesis during both pygidial regeneration and CNS patterning during posterior elongation" are therefore incorrect as this has not been definitely shown by the set of experiments in the paper.

As a similar comment is raised by Referee 2 “The authors should [...] also check in their RNAseq data if other pathways are affected. The authors should also include a more detailed explanation of how the inhibitor acts and what are its potential targets.”,

Below, we propose a common answer to both comments, to avoid redundancy in the letter:

We thank the referees for raising this important point. We strongly believe that studying unconventional model species is a key strategy to assess the diversity, evolution and origin of major developmental processes such as tissue regeneration, or nervous system formation. However, the technical tools available for such organisms are much less developed than in widely used models of developmental biology. To date, there is no means to perform a direct inhibition of Notch or Notch components in regenerating Platynereis worms. Unfortunately, it is not technically possible to perform an inducible genetic knockout that would specifically target a key member of the pathway. The only available approach for functionally studying the Notch pathway in this species is the use of chemical inhibitors, which are widely used for this purpose in non-conventional model organisms such as the cnidarian Nematostella vectensis, the ctenophore Mnemiopsis leidyi, or the placozoa Trichoplax adhaerens (see detailed references below).

That being said, we acknowledge the comments of the reviewers and editor regarding this potential specificity issue, and have taken various steps to strengthen our results.

1/ As requested by Referee 1, we paid specific attention to an alternative key substrate of the γ -secretase beyond Notch: the DCC/Netrin receptor pathway which is particularly involved in axon guidance during later stages of neurogenesis. Based on bibliographic research, we identified more than 25 putative targets genes of this pathway triggered by DCC cleavage (see list below and Table EV6). We identified the homologous genes in Platynereis (identifiers are from Paré et al., 2023) and found none of them to be differentially expressed in our RNA-seq data performed in the presence versus absence of the γ -secretase inhibitor, meaning that DCC/Netrin-dependent genes expressions are not affected by LY-411575 in our model.

Putative target genes / substrates of γ-secretase	Function	Reference(s)	Platynereis ID	DEG_1dpa	DEG_2dpa
DCC	Netrin receptor, axon guidance, neuron survival/death	10.1016/j.pbiomolbio.2015.04.001; 10.1038/s41418-022-01091-z	1717	No	No
Neogenin	Netrin receptor, axon guidance, neuron survival	10.1128/MCB.02114-07; 10.1016/j.ydbio.2006.06.018			

DSCAM	Netrin receptor	10.1016/j.semcd.2020.05.019	3336	No	No
Netrin	netrin-1 (neuron migration); netrin-4 (neuron maturation)	10.1016/j.pbiomolbio.2015.04.001; 10.3389/fcell.2020.590009	11161	No	No
UNC5-A	Netrin receptor; regulate axon guidance	https://www.sciencedirect.com/topics/neuroscience/unc-5	4478	No	No
UNC5-C	Netrin receptor	10.1126/sciadv.abe4499			
Commissureless	axon guidance	10.1016/j.neuron.2015.08.006	No orthologous gene		
Src	Src family kinases (SFKs), downstream of Netrin-1 signaling via DCC	10.1242/dev.044529	14379	No	No
Fyn	Src family kinases (SFKs), downstream of Netrin-1 signaling via DCC	10.1242/dev.044529			
Lck	Src family kinases (SFKs), downstream of Netrin-1 signaling via DCC	10.1242/dev.044529	15365	No	No
Fibronectin (FN1)	cell adhesion, cell motility	10.1038/s41598-019-40886-y; 10.18632/oncotarget.19969	23263	No	No
integrin alpha-2	cell adhesion (receptor for laminin, collagen, collagen C-propeptides, fibronectin and E-cadherin)	10.1038/s41598-019-40886-y	3750	No	No
integrin alpha-5	receptor for fibronectin and fibrinogen	10.1038/s41598-019-40886-y			
integrin beta-1	cell adhesion	10.1038/s41598-019-40886-y	7202	No	No
L1 cell adhesion molecule (L1CAM)	cell adhesion, transmembrane signals, neuronal migration, axonal growth and fasciculation, and synaptogenesis.	10.1038/s41598-019-40886-y; 10.4161/cam.20832; 10.1523/JNEUROSCI.18-10-03749.1998	2063	No	No

ANXA1	inflammatory response	10.1038/s41598-019-40886-y	13813	No	No
NOX1	production of ROS	10.1038/s41598-019-40886-y	12682	No	No
RhoA	Rho GTPases	10.1083/jcb.200405053	49348	No	No
Rac1	Rho GTPases	10.1083/jcb.200405053	49630	No	No
Cdc42	Rho GTPases	10.1083/jcb.200405053	49909	No	No
Nck1	adaptor of Rho GTPases	10.1083/jcb.200405053	20926	No	No
Trio	guanine nucleotide exchange factors (GEFs)	10.1242/dev.044529	456	No	No
Dock1	guanine nucleotide exchange factors (GEFs)	10.1242/dev.044529	845	No	No
Pak1	serine/threonine kinase	10.1242/dev.044529	15018	No	No
ENA/VASP	actin-binding protein	10.1242/dev.044529	9594	No	No
N-WASP	actin-binding protein	10.1242/dev.044529	13674	No	No

Table EV6: Putative members and targets of the DCC/Netrin pathway

Name of the putative target genes, their functions and associated references are indicated. Proteins that are substrates of γ -secretase are indicated in purple. Homologous genes in *Platynereis* were identified, if they exist (identifiers are from Paré et al., 2023). None of them are differentially expressed (or DEG) at 1 or 2 dpa. dpa = days post amputation.

2/ To further extend our analyses on the DCC/netrin pathway, we decided to test an inhibitor, named PP2, of the Src family kinases (SFKs), which are downstream of Netrin-1 signaling via DCC. Upon treatment with 1 μ M of PP2 (lethal at higher concentrations) during posterior regeneration, the process is delayed (barely reaching stage 2.5 at 5 dpa) and the morphology of the regenerated structures, while relatively variable, show important defects at 5 dpa (See figure below = Appendix Figure S7). Detailed comparative analyses of the acetylated tubulin labelling obtained for worms treated with PP2 and LY-411575 at 5 dpa indicated that the phenotypes obtained are not similar.

As expected from the literature, PP2 treatments trigger major axon guidance defects, especially between the ventral nerve cord (VNC) and the gut nerve net (asterisks), that are not observed neither in LY-411575-treated regenerates nor in the controls. The regenerated pygidial nerve organization appears to be affected in some of the samples (see E and F). Besides, it appears that PP2 treatments do not prevent the formation of tissues by the regenerated growth zone, although this aspect is variable at the concentration used. In addition, no particular hypertrophy of the pygidium associated with a dramatic increase in pygidial nervous projections ending up in thick acetylated-tubulin+ foci in the most posterior part of the regenerates were observed, in contrast to LY-411575-treated worms (encircled in white dashed lines). Finally, no thickening of the neurectoderm and VNC were observed in PP2-treated worms. All in all, PP2 treatments trigger specific defects (ectopic nerve projections between the VNC and the gut nerve net, disorganization of the pygidial nerve) that do not correspond to the hallmarks of LY-411575 treatments. Thus, pharmacological

inhibition of DCC/Netrin pathway by PP2 via SFKs does not mimic the phenotype obtained with LY-411575.

To conclude, thanks to these two lines of arguments, we can be confident that the major defects observed upon LY-411575 treatment are not likely due to the inhibition of DCC/Netrin pathway.

Appendix Figure S7: Inhibiting Src family kinases induces a misshapen axon guidance phenotype that differs from the LY-411575 phenotype. A) Violin plots representing the regeneration stages reached by each worm at 2, 3 and 5 dpa during treatment with the inhibitor of the Src family kinases, PP2 at 1 µM in comparison to control (DMSO 0.05%). B to F'''). Acetylated tubulin immunolabelling on whole-mount regenerated parts of DMSO control (B to B'''), LY-411575 (C to C''') and PP2 -treated worms (D to F''') at 5 dpa. Ventral views are on top for each condition and corresponding virtual transverse sections (along (') and ('') in yellow) and sagittal section (('') in red) are at the bottom. In all relevant panels, solid white lines delineate the outlines of the samples, white dashed lines correspond to the amputation planes and white dotted lines delineate the gut. White brackets = ventral nerve cord; white arrowheads = circular pygidial nerve; white double arrowheads = nerves of the pygidial cirri; white dashed ellipse = thick acetylated-tubulin+ foci; white asterisks = ectopic nerve projections between the VNC and the gut nerve net. dpa = day(s) post-amputation. Scale bars = 50 µm. Anteroposterior (A/P) and dorsoventral (D/V) axes are represented. For data presented in A, unpaired Mann-Whitney U tests were used for statistical analyses. P-values are indicated.

3/ In addition, as requested by the Referee 2, we investigated our RNA-seq data and found that relatively few classical signaling pathways are affected by the treatment. They represent only 6 % of the DEG (see Figure 3, Table EV3 and 4). One major pathway that stands out in our analyses, in addition to Notch itself, is the MAPK pathway (Fos, Jun and Maf), consistent with previously described interactions between Notch, API and MAPK signalling pathways in certain types of cancers and whose members are not theoretically major substrate of gamma secretase (Güner and Lichtenthaler, 2020).

While we cannot completely exclude that in addition to Notch, other pathways may be also slightly affected by treatments with the LY-411575 γ -secretase inhibitor, our comprehensive analysis of the most relevant alternative pathway demonstrates that our findings are most consistent with effects on Notch signaling. We have revised our manuscript to reflect the inherent limitations of pharmacological approaches while maintaining confidence in our conclusions based on this additional evidence as follows:

- *We have toned down some of our statements (as requested by Referee 1), especially regarding axon guidance.*
- *We have added a detailed explanation in the Material and Methods of how the inhibitor acts and what are its potential additional targets beyond Notch (as requested by Referee 2).*
- *We have included a "limitation of the study" section after the discussion to develop this point. This paragraph includes appendix Figure S7 and Table EV6, and summarize the supporting elements and new data provided in this letter.*

2. The authors state, based on the data in Figure 4, that "these findings demonstrate that Notch signaling acts as a key regulator of pygidial neurogenesis by controlling neural progenitor specification". This is not well supported by the data. Even if we assume that the inhibitor is specific to Notch signalling, it cannot be deduced from the data what is the proximal cause of the observed phenotypes. Notch signalling could be playing a role in neural progenitors but equally any of the other tissue-level defects that have been described earlier could be secondarily leading to the effects on the nervous system.

The tissue-level defects mentioned and described in Figure EV2 such as a malfunctioning growth zone and segmentation issues are all due to problems with the regeneration of the growth zone stem cells that impact morphogenetic events during subsequent posterior elongation. During posterior regeneration in Platynereis, only the pygidium and the growth zone are regenerated. The newly regenerated growth zone then produces new tissues that will later be segmented and from which appendages (parapodia) will be formed. Segments and parapodia are thus not regenerated per se, they are produced by a regenerated growth zone during post-regenerative posterior elongation.

In contrast, the regenerated pygidium is totally independent of the growth zone tissue production. This structure, which contains muscles, and pygidial cavity does not harbour major tissue-level defects, excepted for the nervous system.

Hence, the tissue-level defects observed in the newly produced tissues by posterior elongation is most probably not the proximal cause of the observed excess of neurogenesis in the pygidium upon γ -secretase inhibition. Tissue-restricted or lineage-restricted Notch perturbations would be necessary to firmly rule out this much less likely interpretation of our data, which is unfortunately not technically possible in this species at the moment.

We thus mentioned this alternative interpretation and toned down our conclusion (Page 13).

3. The next section in title "Notch regulates pygidial neurogenesis through the ligand Delta and several Hes target genes". In this case there is again no evidence on the roles of Delta and Hes genes. The authors show the inhibitors have an effect on these genes but whether the observed effects are "through" the activities of these genes is not explored at all in the manuscript. This would require a lot of functional work on Delta and Hes genes.

We agree with the referee and have changed the title of the section and the text accordingly (Page 13).

4. In their experiments to study the later effects of inhibitors the authors do not discuss the rather larger tissue level differences between control and treated animals. A more thorough explanation of this would be welcome

As mentioned in the last section of the results, Delta, Hes2 and Nrarp expressions in the growth zone are lost in LY-411575 treated worms (treatment from 3 to 10 dpa, Appendix

Figure S6), supporting the idea that Notch regulates not only its regeneration but also partly its molecular identity. This may explain the smaller size of the LY-411575-treated regenerated parts. We also noticed that appendages or parapodia formation is affected. If this is directly linked to Notch inhibition or an indirect consequence of a misshaped peripheral nervous system is unknown and out of the scope of the article.

In addition to the already described defects in the nervous system and chaetal sacs, we have performed additional ISH for markers of the growth zone (Hox 3), progenitors (Nanos), appendages (Dlx) and muscles (Troponin I) (Appendix Figure S6). We have determined that these structures are present and, albeit smaller, look normal, with the exception of the pygidium (labelled by Cdx) which is enlarged. We have added these new expression data and modified the text accordingly (Page 18).

Minor comments:

1. In figure 2, the Authors state that there is no difference in the "distribution of Edu+ cells" but to me the images shown show some drastic differences in the localization of the Edu+ cells, most strikingly in the bottom of Panel 2 where the side with Edu+ cells seems to have completely switched.

We thank the referee for pointing this. We have produced new samples, and have performed EdU labelling, imaging and counting. We have found no difference in the proportions of EdU + cells in this new batch of samples, in comparison to the previous ones. Updated datasets are included in the new version of Figure 2. We realized that, unfortunately, the image in the previous version of Figure 2 (LY-411575, at 5 dpa, transverse section) was inverted (dorsal was up). We corrected this and a new representative image is provided for 5 dpa LY-411575 samples in the updated version of the Figure 2. In addition, we identified differences in the localization of EdU+ cells between control and treated samples at 5 dpa (but not at 2 dpa) between the ectoderm and meso-endoderm. We made quantifications and found significantly less proliferative cells in the meso-endoderm of the 5 dpa LY-411575 samples, in comparison to the controls. These new data are included in the Appendix Figure S2D and the text (Page 8) was modified accordingly.

The same is true in Supp. Fig2D. (now Appendix Figure S2E)

If we indeed found differences in EdU+ cells localization (ectoderm versus meso-endoderm) in the Figure 2, we do not think it is the case for the Appendix Figure S2E (former Supp. Fig2D). In this experiment, the EdU pulse is made at 2 dpa, followed by a chase of 3 days, to determine if cells proliferate and are not blocked after replication. Punctuated diluted EdU signal is visible both in surface and more intern tissues confirming that cells do divide. In contrast, in the Figure 2, the EdU pulse is made directly at 5 dpa (no chase, no dilution of EdU signal). In the Appendix Figure S2E, the diminished signal intensity in the dorsal side, in the treated condition, is due to difficulties to image highly diluted EdU signal (due to this long chase) for thick samples (conversely to the thinner controls) from the opposite ventral side. We have enhanced the EdU signal in the new version of the Appendix Figure S2 to reduce this visual artifact.

The authors should attempt to quantify the position of cells before stating that there is no difference in distribution.

As requested by the referee, we quantified EdU+ cells in the ectoderm versus meso-endoderm at 5 dpa for control and LY-411575 treated worms. As stated above, less proliferative cells are found in the meso-endoderm of LY-411575 treated worms in comparison to controls. Those new data are included in the Appendix Figure S2D and the text (Page 8) was modified accordingly.

2. Based on Supp Fig 3K-M the authors state that "proper gut and muscle regeneration" but the images suggest that it is rather abnormal and perhaps "proper" is not the correct term.

We agree and remove the term "proper".

3. In fig. 3A the multiple colours are confusing. It would be better to make just red genes that are "DEGs" and the rest all grey.

Done

It is not clear in the figure what the gene names point to...i.e. they are not assigned to a particular point in the plot. I would either put them assigned to a point or just mention the names in the text.

Done

4. The numbers in Fig.3A' do not add up. Based on the venn diagram there are 698 genes differentially expressed (313+385) at 1dpa but this is less than shown above. The same for 2dpa genes.

Thank you for noticing this issue. We have checked the numbers and corrected them.

5. In figure 6, the DMSO and treated animals are not labelled like in other figures.

Done

Referee #2:

Summary & Significance

This valuable manuscript of Bideau et al. studies Notch pathway-dependent pygidium regeneration in the juvenile stage of the marine annelid *Platynereis dumerilii*. The work focuses on amputation-induced regeneration, neurogenesis and morphological and transcriptional comparisons in control animals and animals with inhibited Notch signalling. The authors concluded that Notch signalling has multiple conserved functions in juvenile *Platynereis* neurogenesis. The authors present strong evidence to support their conclusion that Notch signalling regulates neurogenesis during posterior regeneration although some concerns remain about the specificity of the inhibitor treatments. The study will be of interest to developmental biologists working on Notch signalling and regeneration.

Strengths

The authors use a clear tail regeneration paradigm and combine it with extensive gene expression analyses and inhibitor treatments. They analyse several Notch signalling-related genes both by high-quality in situ hybridisations and RNA-seq approaches at different stages of regeneration. The authors also quantify cell proliferation, cell death and nervous system morphology. The data are in general of high quality.

We thank the referee for this very positive appreciation of our work. We have addressed all the comments raised, as detailed below.

Weaknesses :

The authors mostly used LY-411575 inhibition to block the Notch pathway. This compound is a very potent γ -secretase inhibitor that inhibits Notch cleavage at low nanomolar concentrations. The authors used 5 micromolar concentration in their experiments. The inhibitor is known to also affect other pathways that depend on γ -secretase activity and at the high concentrations used other side effects may also occur. The authors should carefully consider and discuss these potential side effect and also check in their RNAseq data if other pathways are affected. The authors should also include a more detailed explanation of how the inhibitor acts and what are its potential targets.

*While LY-411575 is indeed a potent γ -secretase inhibitor, we respectfully disagree with the referee's assessment of its efficiency at low nanomolar concentrations. If this may be true in cell cultures or organoids, in whole animals — especially aquatic and marine animals — the concentration range used is at the micromolar level. Additionally, *Platynereis* juvenile worms have a thick cuticle which often prevents drug penetration. The following table summarizes several major studies in which LY-411575 was used to block the Notch pathway in various model species, along with the concentrations used. These concentrations range from 10 to 50 μ M — far beyond the concentration (5 μ M) we used for *Platynereis* regeneration experiments.*

Model species	LY-411575 concentrations used	References
Zebrafish hair cell	10 and 50 μM	Romero-Carvajal A, et al.. Regeneration of Sensory Hair Cells Requires Localized Interactions between the Notch and Wnt Pathways. Dev Cell . 2015 -10;34(3):267-82. doi: 10.1016/j.devcel.2015.05.025.
Zebrafish liver	10 μM	C. Zhao et al. Regenerative failure of intrahepatic biliary cells in Alagille syndrome rescued by elevated Jagged/Notch/Sox9 signalling, Proc. Natl. Acad. Sci. U.S.A. 2022. 119 (50) e2201097119
Zebrafish brain	50 μM	Lara Dirian et al. Spatial Regionalization and Heterochrony in the Formation of Adult Pallial Neural Stem Cells. Developmental Cell , 2014, doi.org/10.1016/j.devcel.2014.05.012.
Mice	0.30 mg kg ⁻¹	Wu LM, et al. Zeb2 recruits HDAC-NuRD to inhibit Notch and controls Schwann cell differentiation and remyelination. Nat Neurosci . 2016 Aug;19(8):1060-72. doi: 10.1038/nn.4322.
Nematostella vectensis embryo	10–40 μM	Haillet, E., et al. Segregation of endoderm and mesoderm germ layer identities in the diploblast Nematostella vectensis . Nat Commun 16, 7979 (2025). https://doi-org.insb.bib.cnrs.fr/10.1038/s41467-025-63287-4
Nematostella vectensis embryo	40–70 μM	Sanjay Narayanaswamy et al. Notch coordinates self-organization of germ layers and axial polarity in cnidarian gastruloids. bioRxiv 2025.09.09.675057;
Trichoplax adhaerens adult	10 μM	Najle, Sebastián R. et al. Stepwise emergence of the neuronal gene expression program in early animal evolution. Cell , 2023, Volume 186, Issue 21, 4676 - 4693.e29
Mnemiopsis leidyi embryo	25 μM	Brent Foster et al. Notch expression during ctenophore development gives insight into its ancestral function. bioRxiv 2025.09.30.679519

However, thanks to our initial experiment to determine the optimal inhibitor concentration (see Appendix Figure S2A, B, C), we know that the Notch pathway is also inhibited by LY-411575 at 1 μM , albeit with non-homogeneous regenerates morphologies. We performed acetylated tubulin labelling on samples treated with 1 μM of LY-411575 at 5 dpa and obtained the same phenotype (see the figure below for a comparison with Figure 2). However, the morphologies of the samples are much more variable, which makes the interpretation of the data more difficult. This is why we decided to use a concentration of 5 μM .

Overall, we are confident that the concentrations used are not too high, thereby preventing the occurrence of side effects as much as possible.

Additional figure for the point-by-point response: Acetylated tubulin immunolabelling on whole-mount regenerated parts of 1 μ M LY-411575-treated worms (B) and controls (A) at 5 dpa. A, B: Ventral view. A', A'', B', B'': corresponding virtual transverse sections (along the yellow dotted lines (') or ('')). solid white lines delineate the outlines of the samples, white dashed lines correspond to the amputation planes and white dotted lines delineate the gut. White brackets = ventral nerve chord; white arrowheads = circular pygidial nerve; white double arrowheads = nerves of the pygidial cirri. Scale bars = 50 μ m.

That being said, as also pointed out by Referee #1, the inhibitor LY-411575 (like RO-4929097 and DAPT) does block gamma secretase, which may cleave other proteins and substrates in addition to Notch receptor. Regarding this point, please, see our common answer to referees #1 and #2 above.

Minor Comments :

Writing style: The manuscript in some sections lacks clarity and there are inconsistencies in the terminology. A detailed list of suggestions is given below.

We thank the referee for the suggestion regarding the writing style and have followed all of them.

The authors could consider including in situ for Jagged and Presenilin and discuss their expression dynamics after amputation (based on their RNAseq data these genes change their expression and they are part of the pathway)

As suggested by the referee, we performed ISH for Presenilin and included the images in the Figure 1. The expression pattern of this gamma-secretase complex member is broad and diffuse in the whole regenerated structure including the neurectoderm. For consistency's sake, we also studied the expression of Presenilin upon LY-411575 treatment at 2, 5 and 10 dpa. The resulting data are presented in Figures 5 and 8.

Regarding Jagged, our RNAseq data showed that the expression of this gene is extremely low and the variations are not significant (between 0.021 to 0.252 TPM). Genes with a TPM below 1 are considered as not expressed. Nevertheless, we performed ISH on key stages and did not detect any signal, as expected (data not shown). Hence, we did not add new data for Jagged in the revised version of the manuscript.

The authors should include a picture of the full juvenile worm highlighting its growth zone and also a summary of the experimental manipulations (position of cuts) and design and the Notch pathway with its core components.

As suggested by the referee, we have added a new figure (Appendix Figure S1), with a picture of the full worm and including a zoom in the GZ and amputation procedure. We have also provided a schematic representation of the Notch pathway including its core components.

at least once per figure indicate the orientation of the specimens with arrows; at least once per figure include a scale bar;

Done

mark the amputation site whenever possible to have a better reference of the relative anatomy

Done

the methods section should include a data and code availability statement with DOI of the repository

We have added a data availability section, which includes accession number for sequencing data from the ENA repository, link to a public github account containing the code and scripts used as well as the accession number for the BioImage Archive repository.

source data for all plots should be provided

Regarding gene expression levels, all data are available in the Tables EV1,3,4 and 5.

Source data used for the plots in the main, expanded view and appendix supplementary figures are now compiled in a multi sheets excel file (Dataset EV1).

please include a more detailed protocol for TUNEL staining and provide all antibody specifications

As requested, we have added a brief explanation about the TUNEL staining in the methods section (Page 25). No antibody is used in this protocol.

please provide more information about the conventional WMISH probes, including plasmid map and availability or PCR primers, probe length

As requested, we have provided more information about the conventional WMISH probes that we used in the Table EV8. Of note, all the genes used in this study were already published in previous studies and deposited on NCBI where all relevant information are indicated. In the list provided, we indicated their accession numbers, the reference of the study which initially used this gene/probe and the plasmid in which the genes are cloned.

Page5 Line14: briefly explain how the previous RNA dataset was generated (e.g. stages, manipulation)

As requested, we briefly described the previous RNA dataset that we mentioned in pages 5/6.

please indicate the number of individuals that were compared per condition in the amputation and in situ experiments

We performed all experiment on a least five individuals for each condition. We indicated this point on method section.

SupFig7: please mention if the seawater (cyan) also contains 0.05% DMSO as the control condition at the same time points

The control condition contains DMSO 0.05 % from day 0 to day 10. When the treatments are delayed and start at day 1, 2, 3, 4 or 5, the worms are kept in sea water before the LY-411575 treatments. We previously determined that DMSO at such concentration does not affect the regeneration timing and is similar to sea water (Planques et al., 2019).

Language:

P4L4: clarify terminology of cirri, antennae, etc.

To avoid confusion, we have removed the terms cirri and tentacles when referring to the peripheral nervous system. We are only using the term cirri in "pygidial cirri". There is no mention of "antennae" in our manuscript as those are specific sensory head structures.

P4L7: mention that HES genes are members of the bHLH family

Done

P4L22: maintain the listing format (Third, ...) for easier read

Done

P5L2: mention pygidium earlier when introducing regeneration of body parts.

Done

the term "growth zone (GZ)" and "segment addition zone" (SAZ) are used in the literature, please mention this or explain the difference

We have mentioned in the text (Page 4) the alternative term found in the literature (SAZ).

Vesicular acetylcholine transporter -should be VAcHT

Done

"Notch inhibition" should rather be written as Notch-pathway inhibition because it is not Notch that is directly inhibited

Done

P5L22: "Strikingly" - avoid words like "Interestingly" or "Strikingly"

Done

P6L1: which six core members & which four are expressed? mention the core members in the introduction

Done

Fig1: NA here means: "healthy" non-amputated control (Ref35), but in B NA means 15 days after amputation

We used the fully regenerated posterior part, 15 days post amputation (dpa), as a proxy of non-amputated or NA. Indeed, at this stage, the worms have regenerated their growth zone for 12 days, started to produce segments since at least 10 days and the regeneration process is considered as largely over. The newly formed structure is integrated in the non-amputated tissues and cannot be morphologically discriminated. But the cuticle is a bit thinner allowing for in situ hybridization to be performed, which is quite challenging in a non-amputated control. It is also worth to note that the worms considered as NA in our culture condition, could have been also naturally amputated in their culture box, while this is not morphologically visible if this amputation occurred more than 10 days before we collected them. Hence, we consider that 15 dpa is a proxy of the NA condition.

P7L21: Shortly state that worms were treated immediately after amputation.

Done

P7L18: describe the hypertrophied tissue in more detail e.g. more extended in the ventral-dorsal axis (round shape) compared to a flat oval shape in the non treated animals.

We have added the mention of the round shapes of the blastemal lobes, to our initial description (Page 9). We cannot add more details on the general morphology of the blastema structures, as suggested by the reviewer, as these details could vary from one sample to another, especially due to the mounting system during imaging. Similar minor variability is present in the controls.

P9L1: only use "significant" if statistical test were done

Done, we have replaced "significant/ significantly" by "massive / massively" when referring to expression patterns.

P8L18: What does "few tissues" exactly mean?, "hypertrophic proliferative pygidium"

Done, we have corrected "hypertrophic" with "hypertrophied".

The exact number of cells produced by the growth zone in LY-41575 treated worms in comparison to the DMSO control is not determined, but the structure produced is much smaller and not segmented. We clarified this in the manuscript.

P14L23: "Superficially is a highly proliferative Ngn+ expressing layer of neuronal progenitors" - use a more precise term than "superficially" e.g. close to the dorsal ectoderm etc
Done, we have modified "superficially" by "On top of the neurectoderm"

Fig6: switch the position order from Collier and Elav to the order you depicted in Fig4, Elav and Collier.

Done

Fig7: In A the virtual cross sections (' & ') are too wide compared to the ventral view and thereby not in proper scale. (see Panel B).

We agree and this is done on purpose. If we make the virtual transverse and sagittal sections in scale compared to the ventral views, they will be extremely small and nothing will be visible. This situation is reflected by the scale bars of different sizes for each view.

Fig7: The longitudinal and cross sections are contradicting each other as the longitudinal section depicts all three markers to be present up to the top layer of the specimen

We thank the referee for pointing out this error. We have corrected the schematic representations of the Synaptotagmin expression (yellow dots) in the Figure 7, panels A6'' and B6'''. These are now consistent with the transverse sections, as they should be.

SupFig2 (now Appendix Figure S2): InE: n is missing; $\mu\text{m}^2 > \mu\text{m}^2$

We have mentioned the number of samples ($n \geq 8$) for the panel F in the legend of the Appendix Figure S2 (previously SupFig2E), as the n is different per stage, and corrected the μm^2 .

SupFig3 (now Figure EV2): align numbering; K2: pygidium seems to be present at 2 dpa although amputated and gone in K'2;

In L'2 (K'2, in the previous version), The sample is slightly smaller than the other 5 dpa under LY. Sample variability is something we encounter relatively often when working on regeneration and posterior elongation, in comparison to development, in particular when using chemical treatment. However, the pygidium is for sure present, as evidenced by the ring-like Twist expression that is typical of the mesodermal tissues of the pygidium. We corrected the numbering alignment.

SupFig4: the black and white schematic is not 100% clear on first sight, maybe add a small description/color parts of the animal to clarify the difference.

We have added a small description of the samples collected in the legend of the Appendix Figure S3, as follows "For the 0 and 1 dpa samples, no structures were regenerated (only a wound epithelium is reformed at 1 dpa), so we only collected only the hemi-segment abutting the amputation plane. For the 2 dpa samples, we collected both the small bilobate blastema formed and the hemi-segment abutting the amputation plane."

Fig6: add treatment & stage of the depicted images;

Done

use comparable specimen regions for the stainings;

Specimen variability, even in the controls, is classical during processes such as posterior regeneration and posterior elongation. To reduce this variability, we provided pictures showing similar parts of the specimens and modified the figure 6 as requested.

add amputation lines in the in situ stainings.

Done

Add description of blue arrowheads.

Done, please see the modified figure legends

Typos –

"prefiguring longitudinal tracks of different neuron types." tracts

"the addition of newly formed segment at their posterior end" -segments

"regulates both the neural progenitor determination and the differentiated neurons balance in the regenerated pygidium" the determination of neural progenitors and the balance of differentiated neurons

The notation "that Platynereis' genome" etc is a bit odd, it reads better as 'the Platynereis genome' or 'the genome of Platynereis'

"are not merely arrested at approximately stage 2, but rather exhibits" rather exhibit

"the two nerves innervating the anal cirri" these nerves rather contain nerves that come from the anal cirri sensory neurons and run into the nerve cord so better to say the two nerves running from the anal cirri or simply the two nerves of the anal cirri

"that early chemical inhibition of Notch pathway" of the Notch pathway

"Notch signaling controls pygidial neurogenesis during regeneration by regulating Hes genes activity" Hes-gene activity or the activity of Hes genes

"Given the dramatic nervous system defects induced by Notch inhibition during posterior regeneration, we decided to further explore its impact on gene expression thanks to a bulk RNA-seq unbiased approach between LY-411575-treated and control worms at 1 and 2 dpa (Sup. Fig. 4A and B, Supp. Table 2)." reword - thanks to does not work here

"and ectopic neurons formation" ectopic neuron formation

All done

"Importantly, one fifth of DEG are related to nervous system," here specify how many of these are up or down regulated

We have specified the percentage of nervous system related genes that are up regulated for both condition in the manuscript (page 11).

"salt and paper fashion" pepper

"Notch signalling pathway inhibitors treatments" Treatments with Notch signalling pathway inhibitors

"Supp. Fig. 4" Sup. Fig. 4

Done.

Following the EMBO instructions, all the supplementary figures are now either an Expanded View figure or an appendix supplementary figure.

Referee #3:

The manuscript presents a significant contribution to our understanding of Notch signalling in bilaterian neurogenesis. The study provides valuable insights into the conserved role of Notch in regulating neural progenitors and neuron fate in the pygidium. Furthermore, the findings are relevant for the Spiralia community, enhancing our comprehension of both regeneration and neurogenesis during growth processes. The results are novel, significant, and clearly stated. The data presented support the conclusions effectively, and the comprehensive methodology employed adds robustness to the findings. The integration of diverse experimental techniques is particularly commendable.

I would like to recommend the manuscript with minor revision. Only minor issues are raised that will further enhance clarity and precision.

We thank the referee for this very positive appreciation of our work. We have made all the minor suggestions requested, as detailed below.

Minor Suggestions:

Figures:

Consider excluding figure descriptors within the text (e.g., change "Fig.1B - green square bracket" to just "Fig.1B"), as this information is clearly presented in the figure legends.

Done

Ensure that figure panels are consistently labelled in the text (e.g., "Supp. Fig. 1B-e, f").

Following the EMBO instructions, all the figure panels mentioned in the text were modified and are now consistent.

Figure consistency by i) using uniform panel labels (e.g., in Fig. 6, the terms "contr" and "inhibition" should be added), ii) ensuring all arrowheads are clearly visible/ included, and iii) increasing arrowhead size in the main figures is recommended, as well as iv) consistent alphabetical labels (e.g., A, B, C...) across supplementary figures (e.g., S-fig 5).

Done, consistent alphabetical labels across supplementary figures were added.

We have increased the size of arrows as much as possible.

Text:

The manuscript is well written and clear, with thorough and well-conducted experiments.

Ensure consistency in terminology (e.g., "anal" vs. "pygidial cirri").

Done, we have chosen to use the term "pygidial cirri".

In the sentence: "To further characterize this phenotype, we conducted an extensive molecular analysis using a set of markers to label different structures and tissues involved in Platynereis'

posterior regeneration (Fig. 2, Supp. Fig. 3) [28, 29, 36]" - Please clarify the main figure numbers, as they seem currently unclear.

Done, we have rephrased the sentence and referred to Figure 2 G (acetylated tubulin labelling).

Methods:

Please include in the Method part details on how EdU+ labelled cells and Tunnel+ cells were counted in Fig. 2C.

Done, we have added the following sentences in the "Images acquisition, treatments and analyses" section of the Methods.

"Briefly, for each sample, all nuclei positions (Hoechst + cells) were identified as spots with a standardized nucleus diameter of 5 μm . A region of interest (ROI) corresponding to the regenerative part was then manually delineated, using the Hoechst signal and the general morphology of the structure. Then, the spots inside the ROI were sorted along the fluorescent signals of the EdU or/ TUNEL. This procedure allowed us to determine the absolute number of nuclei inside the ROI and, among them, the number of positive nuclei for each signal; hence, we could extract the proportions of EdU+ and TUNEL+ for each sample."

Similarly, specify how the density of the surface area was measured in S-Fig 2. It would be helpful to clarify whether a single plane was analysed.

The whole structure was analysed, not a specific single plane. We have added the following sentences in the "Images acquisition, treatments and analyses" section of the Methods to describe how the cell density of a surface was determined.

"Similarly, for measuring cell density of a surface area, we identified all nuclei positions, manually delineated a ROI and determine the surface of the structure, then counted the spots inside the whole ROI. The density was defined as the number of nuclei per μm^2 for the whole structure."

Dear Dr. Gazave

Thank you for the submission of your revised manuscript to our offices. I am sorry it took us a while to get back to you, but as you can imagine these things tend to move slowly during the holidays period. We have now received the enclosed reports from the referees that were asked to assess it. EMBOR-2025-61755V2 still has minor suggestions that I would like you to incorporate before we can proceed with the official acceptance of your manuscript. Moreover, I include below a few points raised by our editorial assistance team that must be addressed before we can accept your paper.

I look forward to seeing a new revised version of your manuscript at your earliest convenience.

Sincerely,
Yehu Moran

Yehu Moran
Academic Editor
EMBO Reports

comments by editorial assistance team (need to be corrected, but not mentioned in the point-by-point letter)

Keywords: missing, please provide.

Conflict of interest/Disclosure and Competing Interests Statement: included, but it needs to be renamed to Disclosure and Competing Interests Statement

REFERENCES: Not meeting our format - need to be alphabetical, not numerical; et al needs to be used after 10 author names; the section heading should be References instead of Bibliography. Please correct.

FUNDING INFO: missing in our submission system - ANR-11-IDEX-0005-01, Université Paris Cité, ENS Lyon, "Investissements d'Avenir" program managed by the Agence Nationale de la Recherche (contract ANR-10-INBS-0009). Please correct.

DATASET EV LEGENDS: there are 8 EV tables uploaded; EV tables can roughly fit an A4 page and can be easily converted to PDF, therefore, the following tables can remain as EV Tables, but they should have their legend on the same tab where the table is so that the files can be properly converted to PDF: Table EV2, Table EV6, Table EV8; so we would have in the end Table EV1-EV3; Tables EV1, EV3, EV4, EV5, EV7 are datasets as they are more complex (have a lot of rows, columns, sheets and cannot be properly converted to PDF) and these should be renamed to Dataset EV1-EV5; these changes should be performed in all places: source file names, legends, titles in the system, callouts in the ms; Dataset EV1 appears to be SD, if yes, then this shouldn't be labeled as such, but as Figure Source data.

SYNOPSIS IMAGE: included, but looks like cover art, needs to be schematic, a sketch of the major findings (not containing data images) - in jpeg, TIFF or png format: 550 pixels wide x 200-600 pixels high. You can look on recently published papers in EMBO Reports for examples.

SYNOPSIS TEXT: missing, please provide following our specific format.

R&T TABLE: included in the manuscript, but it needs to be removed and uploaded separately.

SOURCE DATA: completed SD checklist provided; the authors deposited microscopy images online, please verify that everything is there; Dataset EV1 has the numerical source data but the SD provided in our system need to be uploaded using file type Figure Source Data (not Data set) and the authors should upload them as one zip folder per figure, inside each folder, the files should be organized in subfolders, one subfolder for each panel.

Additional notes:

- The manuscript sections should be in the following order: Title page - Abstract & Keywords - Introduction - Results - Discussion - Methods - Data Availability - Acknowledgments - Disclosure Statement & Competing Interests - References - Figure Legends - (Main Tables with legends if applicable) - Expanded View Figure Legends.
- We received bounced email alerts for co-author Alexandre Couetoux - alexandre.couetoux@universite-paris-saclay.fr. The author should either remove the author from the author list in the system and then add him back using the new email address or send us the new email address and we will update the account accordingly

*Please note that the specific URLs for PRJEB63219, S-BIAD2428 datasets are not provided in the data availability statement. This should be fixed.

Figure Legends - Comments

- Please note that the exact p values are not provided in the legends of figures 2A, EV4 B. Please fix.
- Please indicate the statistical test used for data analysis in the legend of figure 3A.

comments by referees (need to be corrected AND mentioned in the point-by-point response letter)

Referee #1:

Overall, I think the authors have made a very good attempt to address my criticisms and comments as far as they reasonably can. My only remaining concern is that the paper still relies exclusively on γ secretase inhibition, and the additional analyses do not truly resolve this limitation. In fact, I feel the authors now somewhat downplay the possibility of alternative targets too strongly.

In the limitations section, the authors state:

"To support the Notch specificity of the phenotype induced by our GSI inhibition, we first confirmed that downstream genes of the DCC/Netrin pathway were not affected upon GSI treatment in our differential transcriptome (Table EV6). Next, chemical inhibition of Src family kinases (SFKs), a gene family known as effectors of Netrin/DCC pathway, did not produce the same phenotype as the one obtained using GSI (see Appendix Fig. S7). Both these elements support a predominant role of Notch signaling in the observed defects, although potential minor off-target effects of GSI cannot be fully excluded. Similarly, we cannot definitely rule out the possibility that in addition to Notch, other pathways might also be slightly altered by treatments with GSI."

While these analyses suggest that DCC/Netrin signaling and/or SFKs are unlikely to explain the observed phenotype, they do not directly rule out their involvement. More importantly, however, they do not provide strong evidence that Notch is the primary or relevant target. The statement, "Both these elements support a predominant role of Notch signaling," and similar phrasing throughout the manuscript are, in my view, misleading. The authors appear to construct a limited set of alternative pathways and, by excluding them, imply that Notch must therefore be responsible. This resembles a straw-man argument, where the lack of effects on these specific pathways is taken to suggest that Notch is the only plausible target.

In addition, the phrasing "although potential minor off-target effects of GSI cannot be fully excluded" is problematic. The inhibitors used are γ -secretase inhibitors, meaning they directly target γ -secretase itself. γ -Secretase has multiple substrates beyond Notch, and effects on these substrates should not be framed as "off-target" effects. Rather, they represent biologically relevant, on-target consequences of γ -secretase inhibition. As such, it is entirely plausible that the observed phenotype could be mediated through pathways independent of Notch signaling.

I therefore think the authors should revise the limitations section to more honestly reflect this uncertainty. Specifically, they should avoid downplaying the possibility that the phenotype is Notch-independent and acknowledge more clearly that their experimental approach does not allow them to attribute the effects specifically to Notch signaling. A more balanced discussion would significantly strengthen the rigor and transparency of the manuscript

Referee #2:

The authors addressed all my comments.

Regarding the code, I recommend to make a permanent version available on Zenodo and integrate it with the github repo, then add the DOI of this Zenodo record to the reference list.

<https://help.zenodo.org/docs/github/>

GitHub is not a permanent repository for sharing code.

Referee #3:

The manuscript gives a detailed report about the importance of Notch signalling for posterior generation and nervous system development in amputated Platynereis. The text and figures are clear, supporting the authors' claim. It is suitable for publication.

The only point that was noticed is that the units are sometimes written with a space after the number and sometimes without. It may be worth standardising this throughout the text and figures before publication.

Referee #1:

Overall, I think the authors have made a very good attempt to address my criticisms and comments as far as they reasonably can. My only remaining concern is that the paper still relies exclusively on γ secretase inhibition, and the additional analyses do not truly resolve this limitation. In fact, I feel the authors now somewhat downplay the possibility of alternative targets too strongly.

In the limitations section, the authors state:

"To support the Notch specificity of the phenotype induced by our GSI inhibition, we first confirmed that downstream genes of the DCC/Netrin pathway were not affected upon GSI treatment in our differential transcriptome (Table EV6). Next, chemical inhibition of Src family kinases (SFKs), a gene family known as effectors of Netrin/DCC pathway, did not produce the same phenotype as the one obtained using GSI (see Appendix Fig. S7). Both these elements support a predominant role of Notch signaling in the observed defects, although potential minor off-target effects of GSI cannot be fully excluded. Similarly, we cannot definitely rule out the possibility that in addition to Notch, other pathways might also be slightly altered by treatments with GSI."

While these analyses suggest that DCC/Netrin signaling and/or SFKs are unlikely to explain the observed phenotype, they do not directly rule out their involvement. More importantly, however, they do not provide strong evidence that Notch is the primary or relevant target. The statement, "Both these elements support a predominant role of Notch signaling," and similar phrasing throughout the manuscript are, in my view, misleading. The authors appear to construct a limited set of alternative pathways and, by excluding them, imply that Notch must therefore be responsible. This resembles a straw-man argument, where the lack of effects on these specific pathways is taken to suggest that Notch is the only plausible target.

In addition, the phrasing "although potential minor off-target effects of GSI cannot be fully excluded" is problematic. The inhibitors used are γ -secretase inhibitors, meaning they directly target γ -secretase itself. γ -Secretase has multiple substrates beyond Notch, and effects on these substrates should not be framed as "off-target" effects. Rather, they represent biologically relevant, on-target consequences of γ -secretase inhibition. As such, it is entirely plausible that the observed phenotype could be mediated through pathways independent of Notch signaling.

I therefore think the authors should revise the limitations section to more honestly reflect this uncertainty. Specifically, they should avoid downplaying the possibility that the phenotype is Notch-independent and acknowledge more clearly that their experimental approach does not allow them to attribute the effects specifically to Notch signaling. A more balanced discussion would significantly strengthen the rigor and transparency of the manuscript

Response: We thank the referee for his comment. We took into account his concern and modified the "limitation of the study" section accordingly. We clearly state that other targets of GSI could be also partly responsible of the phenotype observed, in addition to Notch, as our experiment approach does not allow to exclude this possibility.

Referee #2:

The authors addressed all my comments.

Regarding the code, I recommend to make a permanent version available on Zenodo and integrate it

with the github repo, then add the DOI of this Zenodo record to the reference list.

<https://help.zenodo.org/docs/github/>

GitHub is not a permanent repository for sharing code.

Response: As requested by the referee, we made the code available on Zenodo and provide the associated doi ([DOI 10.5281/zenodo.18341641](https://doi.org/10.5281/zenodo.18341641)) in the revised version of the manuscript.

Referee #3:

The manuscript gives a detailed report about the importance of Notch signalling for posterior generation and nervous system development in amputated Platynereis. The text and figures are clear, supporting the authors' claim. It is suitable for publication.

The only point that was noticed is that the units are sometimes written with a space after the number and sometimes without. It may be worth standardising this throughout the text and figures before publication.

Response: Done

Dr. Eve Gazave
Universite Paris Cite, CNRS, Institut Jacques Monod, F-75013 Paris, France
France

Dear Dr. Gazave,

I am very pleased to accept your manuscript for publication in the next available issue of EMBO reports. Thank you for your contribution to our journal.

You may qualify for financial assistance for your publication charges - either via a Springer Nature fully open access agreement or an EMBO initiative. Check your eligibility: <https://link.springer.com/journal/44319/how-to-publish-with-us>

Yours sincerely,

Yehu Moran
Academic Editor
EMBO Reports

>>> Please note that it is EMBO Reports policy for the transcript of the editorial process (containing referee reports and your response letter) to be published as an online supplement to each paper. If you do NOT want this, you will need to inform the Editorial Office via email immediately. More information is available here: <https://link.springer.com/partners/embo-press/editorial-policies#Peer%20review>